# A Distinct *Hibiscus sabdariffa* Extract Prevents Iron Neurotoxicity, a Driver of Multiple Sclerosis Pathology

**DOI:** 10.3390/cells11030440

**Published:** 2022-01-27

**Authors:** Manoj Kumar Mishra, Jianxiong Wang, Reza Mirzaei, Rigel Chan, Helvira Melo, Ping Zhang, Chang-Chun Ling, Aldo Bruccoleri, Lin Tang, V. Wee Yong

**Affiliations:** 1Hotchkiss Brain Institute, Department of Clinical Neurosciences, University of Calgary, Calgary, AB T2N 4N1, Canada; mkmishra@ucalgary.ca (M.K.M.); jiwang@ucalgary.ca (J.W.); reza.mirzaei@ucalgary.ca (R.M.); rigel.chan2@ucalgary.ca (R.C.); francisca.cavalcante@ucalgary.ca (H.M.); 2Department of Chemistry, University of Calgary, Calgary, AB T2N 4N1, Canada; pzhang05@gmail.com (P.Z.); ccling@ucalgary.ca (C.-C.L.); 3AnthoBio, Calgary, AB T2N 4N1, Canada; aldo.bruccoleri1@ucalgary.ca (A.B.); linaileentang@gmail.com (L.T.)

**Keywords:** experimental autoimmune encephalomyelitis, human neurons, iron, multiple sclerosis, neurodegeneration, neuroprotection, oxidative stress

## Abstract

Iron deposition in the brain begins early in multiple sclerosis (MS) and continues unabated. Ferrous iron is toxic to neurons, yet the therapies used in MS do not counter iron neurotoxicity. Extracts of *Hibiscus sabdariffa* (HS) are used in many cultures for medicinal purposes. We collected a distinct HS extract and found that it abolished the killing of neurons by iron in culture; medications used in MS were ineffective when similarly tested. Neuroprotection by HS was not due to iron chelation or anthocyanin content. In free radical scavenging assays, HS was equipotent to alpha lipoic acid, an anti-oxidant being tested in MS. However, alpha lipoic acid was only modestly protective against iron-mediated killing. Moreover, a subfraction of HS without radical scavenging activity negated iron toxicity, whereas a commercial hibiscus preparation with anti-oxidant activity could not. The idea that HS might have altered properties within neurons to confer neuroprotection is supported by its amelioration of toxicity caused by other toxins: beta-amyloid, rotenone and staurosporine. Finally, in a mouse model of MS, HS reduced disability scores and ameliorated the loss of axons in the spinal cord. HS holds therapeutic potential to counter iron neurotoxicity, an unmet need that drives the progression of disability in MS.

## 1. Introduction

Multiple sclerosis (MS) is a disorder of the central nervous system (CNS) characterized by the loss of myelin and oligodendrocytes and of neurons and their axons. Symptoms include fatigue, tremor, stiffness, painful spasms, diplopia, poor balance and cognitive impairments, and these vary from one patient to the next depending on where lesions occur in the CNS [1]. In most patients, MS begins with discrete episodes of neurological dysfunction (relapses) followed by remission; this is relapsing–remitting MS [2]. Within 10–15 years in many individuals, there is disability progression often independent of relapse(s), and this is referred to as secondary progressive MS. In a minority (10–15%) of patients, disability progression is unrelenting from the outset, referred to as primary progressive MS. Before the formal diagnosis of MS, a first demyelinating episode suggestive of future MS is termed clinically isolated syndrome. Radiologically isolated syndrome refers to asymptomatic individuals who have undergone a brain magnetic resonance imaging (MRI) scan for various reasons, and where an MRI lesion with similarity to those characterized in MS is observed. Emphasizing the need to treat MS early, brain atrophy is already observed in radiologically isolated syndrome [3], where ~50% of subjects are destined to be diagnosed with MS within 10 years [4]. 

While the initiating cause of MS is unknown, immune cells are activated in the periphery and in the CNS; several peripheral blood leukocyte populations enter the CNS to produce neural degeneration [5,6]. In addition to mediators of injury produced by immune cells, such as cytokines, proteases and reactive oxygen species, factors released by CNS structures can also be injurious, and one of these is iron [7,8]. 

Iron is a co-factor required for many processes in the CNS, including the formation of myelin by oligodendrocytes [7]. It is also a transition metal, changing between ferrous and ferric forms, with a strong capacity to generate oxidative stress. The reaction of ferrous iron with H_2_O_2_, referred to as the Fenton reaction, produces the highly reactive hydroxyl radical. Iron is thus managed safely in homeostasis by a range of mechanisms, including iron importers and exporters on the surface of cells, by transporters such as transferrin, by storage within cells in ferritin cages, and by a variety of ferroxidases [9]. Inflammatory processes alter several of the regulators of iron handling in cells and exacerbate iron accumulation [10]. If unbuffered, the extracellular ferrous iron becomes a mediator of cell injury. 

The potential dangers of iron in causing neurotoxicity across several neurological diseases including MS, and in normal aging, has been reviewed [11]. Iron damages cells through several mechanisms, including the induction of ferroptosis [12,13] and the generation of reactive oxygen species (ROS) [14,15]. Unbuffered iron can also cause injury by activating sphingolipid metabolism that promotes cell death [16] and by misfolding and aggregating proteins including amyloid beta, tau and alpha-synuclein [17,18,19,20]. In tissue culture, iron is lethal to human neurons [21]. 

Iron accumulation is documented in MS via MRI and the Turnbull histological stain in several brain regions, particularly the deep gray matter important as a conduit between the spinal cord and upper brain regions [22]. The detection of iron via MRI in the deep gray matter is found from the early stages of MS, including clinically isolated syndrome [23,24]. Iron accumulation is progressively elevated with advancing disease [25], is associated with the accrual of disability [26] and corresponds to brain areas with neuronal degeneration and demyelination [27,28,29]. The elevated iron in MS brains is thought to be due to be its liberation from damaged, iron-rich oligodendrocytes and myelin [7,28,29]. 

Despite the potential of iron to be injurious in MS, none of the disease-modifying therapies used in MS are reported to directly neutralize iron neurotoxicity. Oxidative stress caused by ROS is increasingly being targeted in MS, an example of which is the use of the anti-oxidant alpha lipoic acid [30,31], which has been shown to reduce MRI-detected annualized brain atrophy by 68% in a small Phase 2 clinical trial [32]. However, ameliorating ROS produced by iron may not overcome iron toxicity if other mechanisms noted above of iron-induced injury (e.g., misfolding of proteins) are prominent. Effective means to ameliorate iron neurotoxicity are thus required. 

*Hibiscus sabdarrifa* (HS) is a species of the genus Hibiscus native to West Africa, but it also grows in other geographic locations. Its extracts are consumed in many cultures, where its herbal tea is known variably as sorrel, roselle, Jamaica and karkade, amongst others. It is used as a food coloring, and clinical, trial-validated results have shown it can treat hypertension [33]. It is also used for hyperlipidemia and glycemia [34]. 

Organic hibiscus extracts, predominantly of its calyces and enriched in anthocyanins [35], have anti-oxidant [36] and anti-inflammatory [37,38] activities in vitro. One extract protects against serum- and glucose-deprivation injury to a neuronal cell line in culture [39]; another study demonstrated the protection of another neuronal cell line against H2O2 toxicity [40]. Hibiscus extracts have been reported to reduce brain pathology and biochemical indices of oxidative stress when injected intraperitoneally into mice in models of Parkinson’s disease [41,42] and Alzheimer’s disease [43]. In high concentrations (>50 μM of hibiscus delphinidin 3-sambubioside), however, anthocyanin-rich hibiscus can be toxic to cells through mitochondria-induced oxidative stress [44]. 

Here, we isolated an HS extract with low anthocyanin content and tested its efficacy to neutralize iron toxicity in neurons in culture and in reducing axonal degeneration in the EAE model of MS. We described the unexpected observations that HS potently attenuates iron neurotoxicity in culture in conditions in which strong anti-oxidants (ferulic acid) or disease-modifying therapies used in MS (dimethylfumarate, fingolimod and siponimod) could not. We also demonstrated that HS ameliorates experimental autoimmune encephalomyelitis (EAE) disease accompanied by substantial protection against the axonal degeneration of the spinal cord. 

## 2. Materials and Methods

### 2.1. Preparation of HS

Aqueous extracts were obtained from dried calyxes of HS plants. The extraction was performed using a 1:8 to 1:10 (*w*/*w*) ratio of dried calyxes to water, and the extracts were processed by filtration and centrifugation to remove solids. The composition of the complex aqueous extract is only partially characterized at this point, but it has low anthocyanin content compared to a commercially available capsule (see Section 3), several organic acids (e.g., glyceric acid, propionic acid, malic acid and oxalic acid) as determined using mass spectrometry and several polyphenols (e.g., gallic acid, gallocatechin and kampferol) as evaluated using liquid chromatography (data not shown). Intellectual property protection is being applied for regarding the HS compositions.

The HS liquid was dried to a powder form. Specific stock concentrations (15 mg/mL or 150 mg/mL) were then reconstituted in water, and working stocks were further diluted with water or tissue culture medium from the stock concentrations. The pH of these stock solutions was ~2.9. In some experiments, HS solution was subfractionated by passing it through a Diaion HP-20 column, and the results of HS fraction 1 are reported here. The anthocyanin level of HS solutions was measured using the total monomeric anthocyanin content assay following protocols described by others [45]. 

We also compared a commercial hibiscus capsule (product # NPN 80051271) purchased from Nutritional Fundamentals for Health Inc, where the label on the bottle reads ‘Hibiscus SAP, Blood pressure and cardiovascular support’. The dry powder within the capsule was constituted in water at 15 mg/mL or 150 mg/mL stock concentration. 

In summary, in this study, we tested HS as an aqueous extract of the dried calyxes of *Hibiscus sabdariffa* plants, different HS batches to evaluate the reproducibility of extracts to protect against the iron-mediated killing of neurons, an HS fraction 1 to determine whether a particular composition of HS retains its neuroprotective activity and a commercially available *Hibiscus sabdariffa* capsule to investigate whether it could replicate the neuroprotective activity of our HS extracts. 

### 2.2. Culture of Human and Mouse Neurons, and Evaluation of Neurotoxicity

Human fetal brain specimens were obtained from legal abortions, and their use for research following parental consent was approved by the Conjoint Health Research Ethics Board at the University of Calgary. Ethics guidelines preclude access to patient information and the cause of the abortion. Neurons of over 80% purity were obtained as previously described [46,47]. In brief, 0.75–1 × 10^5^ neurons were seeded into each well of poly-L-ornithine-coated, 96-well, flat-bottom black/clear plates (Falcon 353219) in 100 µL of AIM V media (Thermo Fisher, Burlington, ON, Canada). These neurons were then used for experiments after 24 to 48 h of culture. Four replicate wells were used per experimental condition.

Primary mouse cortical neurons from embryonic day 15–16 pups were isolated and nurtured in Neurobasal Plus media (Thermo Fisher) with B27 Plus supplement (Thermo Fisher) as detailed [48]. Neurons were grown in poly-L-ornithine-coated, 96-well, flat-bottom black/clear plates at a density of 0.75 × 10^5^ neurons. 

In experiments involving test reagents (e.g., HS or medications used in MS) and iron, the test reagents were applied to neurons 1 h before the iron. At the end of an experiment, cultures were fixed with 4% paraformaldehyde and then incubated with a Tuj1 primary antibody to label tubulin β3 for neurons [49]. Following a secondary antibody incubation, labeled cells in 96-well, flat-bottom black/clear plates were imaged using the 10x/0.5 NA air or the 20x 0.45 NA air objective on the ImageXpress Micro XLS High-Content Analysis System (Molecular Devices). For each well of cells, 9 or 12 (identical within an experiment) fields of view (Figure 1A) that were in the same locations in each well of every well of an experiment were imaged for quantitative analysis. Multiwavelength cell scoring analysis in the MetaXpress High-Content Image Acquisition and Analysis Software (Molecular Devices) was used to quantify the number of neurons in these FOVs, as described previously [21], thereby providing an index of cell survival. Note that this ImageXpress method to quantify the loss of stained neurons and potential protection by test compounds was used in our previous reports where neurons were destroyed using activated microglia [50], activated T cells [51], oxidized phosphatidylcholines [49] and iron [21,52].

It is important to note that there was variation across different batches of cultures, but that all test conditions were controlled within an experiment. These variations across batches of cells prepared at different times were contributed to by the initial health of the isolated cells (some preparations were healthier than others despite the desire to be as consistent as possible), different individuals isolating the cells in the first place, the density of the cell suspension for plating, the susceptibility of neurons to iron-mediated killing that can vary across preparations for reasons that are unclear, and whether 9 or 12 fields of view were quantitated in ImageXpress. Such variability across batches is reflected in the number of control neurons that differed across experiments (see, for example, the *y*-axis numbers in Figure 1 and Figure 2). However, each experiment was internally controlled to have the same conditions of assessment. Moreover, the trend of each key result was evaluated over 3 different batches of neuronal cultures in separate experiments.

### 2.3. Life Imaging of Neurons 

We sought to visualize the progressive injury to neurons exposed to iron in real time. Mouse neurons in a 96-well plate were incubated with 5 μM of Cell Tracker™ Red CMTPX Dye (Thermo Fisher, catalog number C34552), which crosses the plasma membrane of all cells, and with 5 μM of SYTOX™ Green Ready Flow™ Reagent (Thermo Fisher, catalog number R37168). SYTOX is a high-affinity nuclei acid stain that is excluded from healthy cells but which penetrates cells with compromised plasma membrane. HS (100 μg/mL) was then added to pre-specified wells. One hour later, 50 μM of FeSO_4_ was added to control or HS-exposed cultures. After 8 h in a 37 °C incubator, the plate was taken for live cell imaging using the automated ImageXpress Micro XLS High-Content Analysis System under controlled environmental conditions (37 °C and 5% CO_2_). Images were taken at a prespecified location (center of well) every 30 min from 8 to 20 h (where zero is the time of addition of iron to the iron-containing wells). The images were then exported using the MetaXpressR software of ImageXpress into ImageJ. In ImageJ, images from each well were aligned and collated together chronologically as a video. 

### 2.4. Iron Chelation Analysis

The capacity of HS to directly chelate iron was evaluated using the chrome azurol sulfate spectrophotometric assay, as described by others [53]. EDTA was used as a positive control. The ferrous ion-chelating (FIC) ability (%) was obtained using the equation [(Ac − As) /Ac] × 100, where Ac is the absorbance of the control solution (containing all reagents except for the HS), and As is the absorbance in the presence of the HS (or EDTA) sample. 

### 2.5. HORAC and ORAC Assays

The hydroxyl radical antioxidant capacity (HORAC) and oxygen radical absorbance capacity (ORAC) assays were conducted using kits from Abcam (Toronto, ON, Canada). The catalog number for the HORAC assay kit was ab242299, while that for ORAC was ab233473. Values were displayed as gallic acid equivalent (GAE) and Trolox equivalent (TE), respectively. The assays were performed as detailed in the instructions provided by Abcam. 

### 2.6. Assessment of HS against Other Mediators of Neuronal Death

We evaluated whether HS could protect against non-iron mediators of neuronal death. The exposure of neurons to amyloid beta and rotenone is used in models of Alzheimer’s and Parkinson’s disease, respectively [54,55]. The protein kinase C inhibitor, staurosporine, at high concentrations kills cells through apoptosis [56]. Thus, neurons were exposed to 20 μM of beta-amyloid peptide (1–42) (rPeptide), 10 μM of rotenone (Sigma, Oakville, ON, Canada) or staurosporine (Sigma), with or 100 μg/mL HS that was applied 1 h before the toxins. Cells were harvested 24 h after and stained using a Tuj1 primary antibody as above. 

### 2.7. Proliferation of Splenocytes

Murine splenocytes in 10% fetal bovine serum-supplemented RPMI-1640 medium were labeled with 1 μM of carboxyfluorescein succinimidyl ester (CFSE) dye, and HS was then added to specific groups. Cells (100,000/well) were then added to round-bottom wells in a 96-well plate that were uncoated (i.e., control, inactivated condition) or coated with 1 mg/mL anti-CD3/CD28 antibodies that activated T cells. After 72 h, cells were subjected to flow cytometry (FACS Attune NXT) to measure CFSE dilution, where a more proliferative culture would have more cycles of diluted CFSE. 

### 2.8. EAE and Spinal Cord Histology

All experiments were conducted with ethics approval from the Animal Care Committee at the University of Calgary under guidelines of the Canadian Council of Animal Care. Eight-to-ten-week-old C57BL/6 female mice (Charles River Laboratories, Montreal, QC, Canada) with an average weight of 22 g were immunized subcutaneously with 50 μg/100 μL of myelin oligodendrocyte glycoprotein (MOG) 35–55 peptide (Protein and Nucleic acid facility, Stanford University, USA) in CFA supplemented with 4 mg/mL heat-inactivated Mycobacterium tuberculosis H37Ra (Sigma). This method has been previously described [21,47]. A 50 μL emulsion was deposited on either side of the tail base. Pertussis toxin (PTX) (300 ng/200 μL, List Biological Laboratories, Campbell, CA, USA) was injected intraperitoneally on days 0 and 2 after MOG immunization. Daily monitoring of EAE mice was performed, and the mice were scored on a scale of 0 to 15 [47]. At tissue harvest, mice were killed with ketamine (100 mg/kg) and xylazine (10 mg/kg) intraperitoneally. A total of 15 ml of PBS was then perfused via cardiac puncture followed by perfusion for 15 min of 4% paraformaldehyde in PBS. The spinal cord was removed, and the thoracic cord was collected into 4% paraformaldehyde for fixation overnight. Spinal cords were transferred to 30% sucrose solution for at least 48h and were then frozen in FSC 22 Frozen Section Media (Leica, Buffalo Grove, IL, USA). Using a cryostat (Thermo Fisher), spinal cord tissue was cut longitudinally into 20 µm sections, collected on to Superfrost Plus microscope slides (VWR) and stored at −20 °C prior to analysis. 

For immunofluorescence staining, microscope slides containing spinal cord sections were warmed to room temperature for 10 min. Samples were then rehydrated with PBS for 10 min and permeabilized with 0.2% Triton-X100 for 10 min. Horse blocking solution (PBS, 10% horse serum, 1% BSA, 0.1% cold fish stain gelation, 0.1% Triton X-100, 0.05% Tween-20) was used to block the sample for 1 h at room temperature. An additional step was conducted for MBP staining, where sections were delipidated by a successive wash of 50%, 70%, 90%, 95%, 100%, 95%, 90%, 70% and 50% ethanol prior to rehydration with PBS. Primary antibodies used were rabbit anti-human/mouse IBA1 (1:1000, Wako, Richmond, VA, USA), mouse anti-human/mouse tubulin β3 (1:500, Biolegend, San Diego, CA), rat anti-human CD45 (1:200, ThermoFisher), rat anti-mouse MBP (1:200, Abcam), and rabbit anti-mouse NF-H (Encor Biotechnology, Gainsville, CA, USA). Primary antibodies were applied overnight at 4 °C. After washing, the following secondary antibodies (Jackson ImmunoResearch, Grove, PA, USA) were applied at 1:400 dilution of the commercial stock vial: Alexa Fluor 488 donkey anti-mouse IgM, Alexa Fluor 647 donkey anti-mouse IgM, Alexa Fluor 488 donkey anti-mouse IgG, Cyanine Cy3 donkey anti-rat IgG, Alexa Fluor 488 donkey anti-rabbit IgG and Alexa Fluor 647 donkey anti-rabbit IgG.

Fluorescence images of entire longitudinal sections were acquired on an Olympus VS120 Virtual Slide Scanner with a 20× objective (0.75NA PlanApo aperture) and at fixed camera exposure for each fluorescent probe. All images were processed and analyzed using ImageJ (National Institutes of Health, Bethesda, MD, USA) as reported elsewhere [57]. Images were first converted from 16 bit to 8 bit and split into separate fluorescence channels corresponding to each marker. The polygon tool was used to define the white matter as the region of interest (ROI), while grey matter was cleared and excluded from the analysis. For each channel, a separate intensity threshold was determined, such that positive cells were identified, and applied to segment all images. The segmented images were taken to determine the area fraction of positive cells for each marker. This quantitative method was recently detailed [57]. 

It would be of importance to assess changes to neurotransmitter systems in EAE and to determine whether these are rescued by HS. However, EAE is unpredictable in the location of lesions in the spinal cord, and EAE neuropathology is affected differently from one mouse to the next, so that it would be difficult to accurately and reliably assess the changes to neurotransmitters in an EAE experiment. For this reason, we developed quantitative axonal counts through the white matter of the entire thoracic cord [57], so that changes to axonal perturbation from one mouse to another could be meaningfully captured across a large area of the spinal cord in an experiment. 

### 2.9. Statistical Analysis

Data were collated in Microsoft Excel. Graphs were generated using GraphPad Prism 8 (GraphPad, LaJolla, CA, USA). Data shown are the individual data points, where each point on a bar graph represents a biological replicate (for in vivo experiments) or replicate (for in vitro experiments) and mean +/− SD. A *t*-test was used to compare between 2 groups. Unless otherwise stated, one-way ANOVA with Dunnett’s multiple comparison test was used to analyze statistically significant differences between the means of two or more treatment groups against the control group. Asterisks indicate significance, where * *p* < 0.05, ** *p* < 0.01 and *** *p* < 0.001.

## 3. Results

### 3.1. The Killing of Human Neurons in Culture by Iron Is Attenuated by HS, and This Is Not Due to Direct Iron Chelation

Figure 1A is a schematic of the nine fields of view that were automatically imaged per well to represent the total number of neurons remaining after 24 h in that well. Figure 1B captures a low-magnification image of control neurons, neurons exposed to iron for 24 h, and neurons treated with a representative HS and iron. This provides a visual example that the substantial loss of neurons caused by iron exposure is protected by HS. 

Human neurons are highly susceptible to killing by 50 µM of ferrous iron in culture [21]. Exposure to 50 µM of FeSO_4_ for 24 h resulted in the obvious loss of neurons (Figure 1C). However, in the presence of HS at 75–600 µg/mL, which were concentrations that were empirically chosen to span a range, human neurons survived the iron exposure. Complete neuroprotection was achieved by 150–600 µg/mL HS (Figure 1D). 

We assessed the capacity of HS to directly chelate iron and found no evidence of this capacity even at a very high concentration (3 mg/mL) of HS. In contrast, the positive control EDTA had robust iron chelation activity (Figure 1E). 

### 3.2. Mouse Neurons Are Also Protected by HS against Iron Neurotoxicity in Culture

Because of the limited supply of primary human neurons, we cultured mouse neurons to corroborate the human results. Figure 2A,B demonstrate that mouse neurons in culture were also highly susceptible to iron-induced killing. Seven separate extractions of HS tested at 100 µg/mL each completely neutralized iron neurotoxicity (Figure 2A). 

If administered in vivo, HS is likely to encounter different pHs in the body, for example, a pH of 4 at the upper part of the stomach and a lower pH in the bottom half. Moreover, the pH of blood is about 7.4. Thus, we made HS solutions of different pHs by increasing the pH of stock HS (pH 2.9) to 4, 7 or 10 using drops of 1N NaOH. Moreover, the stock HS (pH 2.9) was heated for 1 h at 90 °C and then allowed to cool to room temperature. These were then tested to determine if they could protect against iron toxicity. Figure 2B,C show that regardless of the pH of the solutions, and despite heating, HS ameliorated iron toxicity on mouse neurons. 

A concentration response of HS was next assessed against 50 or 100 µM of FeSO_4_. Against the lower concentration of iron, HS from 50 to 150 µg/mL completely protected mouse neurons against 50 µM of FeSO_4_. Against the higher 100 µM of FeSO_4_, HS at 50 µg/mL was partially protective, but higher concentrations of HS afforded complete protection (Figure 2D). 

### 3.3. Life Imaging of Neurons 

Appendix A shows a control mouse neuronal culture without HS or FeSO_4_ exposure. From 8–20 h, few of the CMTPX-labeled cells (which marks all cells) were SYTOX (green)-positive. As noted in the Methods, SYTOX only crosses into cells with a disrupted plasma membrane. In contrast, in cultures exposed to FeSO_4_, the number of cells that turned green progressively increased, and this was strikingly prevented by HS. 

### 3.4. HS Is Superior to MS Medications for Neuroprotection against Iron 

Medications used to treat MS are based on their capacity to reduce the dysregulated immune system in MS [58]. However, with the recognition that MS involves the substantial degeneration of neurons and axons that drives the progression of disability [6,8], there has been interest in whether MS medications have direct capacity to protect neurons. In this regard, dimethylfumarate (and also its metabolite monomethylfumarate (MMF), to which dimethylfumarate is rapidly converted in vivo) has been shown in culture to protect against the H_2_O_2_- or amyloid-beta-induced death of neurons through an Nrf2-dependent pathway [59,60], while fingolimod protects neurons against oxidative stress or glutamate excitotoxicity by activating sphingosine-1-phosphate receptors [61,62]. Another drug, laquinimod, reduces brain atrophy in patients with MS [63,64] and has protective effects on neurons against various insults [50,65], although it has not received regulatory approval for use in MS. Whether these could protect against iron neurotoxicity has not been addressed. Thus, we compared the potency of HS against these MS medications.

Human neurons were exposed first to HS, MMF, fingolimod or laquinimod, and this was followed 1 h later by FeSO_4_. Twenty-four hours later, the number of surviving neurons was evaluated. Figure 3A shows that none of the MS medications at concentrations used in the literature protected neurons against iron neurotoxicity, while HS (100 μg/mL) was effective. 

Siponimod is another MS medication with perceived activity within the CNS of MS patients [8]; indeed, it is an approved medication for active secondary progressive MS, which has marked neurodegenerative processes. In tissue culture, siponimod protects neurons against astrocyte-induced toxicity [66]. The intraventricular infusion of siponimod in the EAE model of MS prevents synaptic degeneration and preserves neurons [67]. However, when tested against iron in culture, siponimod was ineffective in conditions in which HS was completely protective (Figure 3B).

### 3.5. Neuroprotection against Iron Is Not Related to Anthocyanin Content

Many plant products have high anthocyanin content, and anthocyanins can have neuroprotective activities [68], although high levels are also toxic [44]. We evaluated the total anthocyanin content of HS and found relatively low levels (3.16 ± 0.28 mg/g, mean ± SEM across eight different HS extracts) compared to the commercial hibiscus capsule (9.04 mg/g). HS fraction 1 had no measurable anthocyanin content. Other differences in chemical constituents might exist between HS fraction 1 and commercial hibiscus, but we did not document these. When evaluated against the iron-mediated killing of neurons, HS fraction 1 completely protected neurons, while the commercial hibiscus (50–100 µg/mL) was devoid of an iron neuroprotection capacity (Figure 3C). 

### 3.6. Neuroprotection against Iron Is Not Proportional to Anti-Oxidant Activity 

We evaluated the capacity of HS to scavenge and prevent hydroxyl radical degradation of fluorescein in the HORAC assay, using gallic acid equivalent (GAE) to inform on the HORAC activity of 10 separate HS preparations. The mean HORAC activity was 575 µmole/g (Table 1), while that of the commercial hibiscus exceeded the standard curve (>800 µmole/g). HS fraction 1 had no measurable HORAC activity. For scavenging of peroxyl radical, using the ORAC assay, HS but not its fraction 1 had activity. The HORAC and ORAC activity thus did not predict the iron neuroprotection, as HS and HS fraction 1 protected neurons but the commercial hibiscus could not (Figure 3B,C and Table 1). 

Further unexpected separation of anti-oxidant activity from iron neuroprotection was evident by ferulic acid, a very strong anti-oxidant [69]. Indeed, ferulic acid had HORAC and ORAC activities that were over 10-fold those of HS (Table 1), but it failed to block the killing by iron of neurons (Figure 3B). 

Finally, we examined alpha lipoic acid [31,70,71], an anti-oxidant being tested in a Phase 2 clinical trial in progressive MS (ClinicalTrials.gov Identifier: NCT03161028), where it reduced brain atrophy in a previous Phase 2 clinical trial of patients with secondary progressive MS [32]. Table 1 shows that alpha lipoic acid was comparable in HORAC and ORAC activities with HS. Notably, while HS completely prevented iron toxicity, alpha lipoic acid at 50 µg/mL was only partially effective (Figure 3B). Lower concentrations of alpha lipoic acid from 1–5 µg/mL did not confer any protection (data not shown). In MS patients taking 1200 mg alpha lipoic acid, the dose in MS clinical trials, the median peak level of alpha lipoic acid in the serum was 4.8 µg/mL [72].

### 3.7. HS Also Protects Neurons against Other Stressors

The mechanism(s) by which HS protects neurons against iron toxicity is unknown, as the above data suggest a separation of anti-oxidant activity from iron neuroprotection in the conditions tested. It is possible that particular HS constituents, alone or in combination, alter properties within neurons that enable the cells to protect against subsequent toxin exposure. To support this hypothesis, we pretreated neurons with HS (100 µg/mL) for 1 h and then exposed them to beta-amyloid (20 µM), rotenone (10 µM) or staurosporine (200 nM). After 24 h, few neurons remained in toxin-exposed cultures, but this loss was attenuated by HS (Figure 4). In comparison, neurons pretreated for 1h with monomethylfumarate or laquinimod did not reduce the toxicity of beta-amyloid, rotenone or staurosporine. Fingolimod did not counter the killing of neurons by rotenone or staurosporine, but it decreased the toxicity of beta-amyloid, as has been reported by others [73].

### 3.8. HS Does Not Affect the Proliferation of T Cells in Culture 

We investigated whether HS affects the activity of T lymphocytes since the over-activation of pro-inflammatory T cells helps drive inflammatory conditions such as MS [74,75], and disease-modifying therapies used in MS have potent effects on T cells [58]. Murine splenocytes were subjected to no activation, or to polyclonal (anti-CD3/anti-CD28) activation, with or without HS treatment. After 72 h, CFSE dilution was measured using flow cytometry, where a more proliferative culture would have more cycles of diluted CFSE. Figure 5 shows that polyclonal activation increased the cycles of CFSE dilution (number of humps to the left of the tall peak), but this was not reduced by HS from 75–300 μg/mL.

### 3.9. HS Reduces the Severity of Murine EAE and Confers Neuroprotection In Vivo

We tested the efficacy of HS (whole HS extract as used in Figure 1) in mice immunized to develop EAE. On the day of immunization, HS was initiated once a day at 250 mg/kg by oral gavage. Seven immunized mice were administered HS, while five mice received vehicle (water). Mice in the vehicle group manifested clinical signs from day 13 and were increasingly sick: initially this manifested as a limp tail, then they additionally acquired hind limb followed by forelimb paresis, which was close to paralysis in some cases (Figure 6A). All five vehicle mice were sick. In mice with HS, five of seven mice did not show EAE clinical signs, while one had breakthrough vehicle-level EAE. The last had mild disease. The average of the disability scores in the HS group amounted to one on a fifteen-point scale, which represents mild tail disability (i.e., no limb involvement) (Figure 6A). Appendix A displays the movement of a representative (based on the average disability score) mouse in the vehicle or HS groups just before the termination of the experiment on day 24. While the vehicle-EAE mouse had tail disability (i.e., not raised from the ground) and obvious hind limb impairment, the HS-EAE appeared normal with the exception of the tail being on the ground. 

Mice were killed at day 24, and the spinal cord was processed for histological assessments of neuropathology. Figure 6B displays examples of stains in longitudinal sections of the spinal cord of EAE mice (vehicle treatment). An active lesion is informed by an aggregate of CD45+ leukocytes, some of which are CD3+ T cells and Iba1+ microglia/macrophages. As CNS lesions in EAE can vary in size and location, we quantified the histological outcomes by evaluating the thoracic spinal cord in longitudinal sections for each mouse. Figure 7A shows examples of longitudinal sections used for quantitation. The images are from a vehicle-treated EAE spinal cord stained for CD45 (immune cells), CD3 (T cells) or Iba1 (microglia/macrophages), or for the degree of myelin (myelin basic protein, MBP+) or axonal (neurofilament heavy chain, NFH) loss. Sections with a central canal were chosen for quantification to normalize location across mice. For quantitation, slide scanner images of both the right and left lateral columns were processed by ImageJ software for the total area per section bordered by the stain of interest, as described recently [57]. Of this, the area and thus the % covered by the thresholded stain was then obtained. In addition to the EAE-vehicle (control) or EAE-HS groups, naïve mice with no EAE induction were also assessed.

Figure 7B shows that the degree of CD3 T-cell neuroinflammation was not different across the groups, corroborating the tissue culture data that HS does not direct affect T-cell proliferation. In contrast, the CD45 immunoreactivity, which encompassed all leukocytes and also microglia, was shown to be reduced by HS compared to control-EAE. 

Finally, we evaluated the extent of myelin and axonal loss in the lateral columns using immunoreactivity for MBP and neurofilament, respectively. While the myelin loss occurring in EAE was not affected by HS (*p* = 0.052 comparing control-EAE versus HS-EAE in the ANOVA Dunnett multiple comparisons), the prominent loss of axons in EAE-control was significantly prevented by HS (Figure 7C). Remarkably, the axonal density in the lateral columns of the thoracic spinal cord in EAE mice treated with HS was similar to that of naïve controls. Thus, HS is neuroprotective against the neuroaxonal loss that occurs after EAE injury.

## 4. Discussion

The neuropathology of MS includes the loss of oligodendrocytes and myelin and of neurons and their axons. The latter neuroaxonal injury and loss correlates with neurologic disability in MS, as evaluated using MRI and postmortem histology studies [76,77], and is thought to drive the progression of disability. Other debilitating symptoms of MS, including cognitive impairments that occur from early in MS, also correlate with brain atrophy [78]. Indeed, much data affirm that the neuroaxonal injury and loss occur earlier in the disease before formal diagnosis. Atrophy is observed in radiologically isolated syndrome (RIS), which is the earliest detection via MRI of possible future MS and where ~50% of subjects are diagnosed with MS within 10 years [4]. Atrophy is observed in RIS in the spinal cord [79], thalamus [80] and cerebellum [3]. For clinically isolated syndrome, which is the first clinically documented demyelinating event and where a second episode would lead to the diagnosis of MS, there is extensive gray matter atrophy in several brain areas [77,81]. Another index of neuroaxonal degeneration is the MR spectroscopy-detected reduction in brain N-acetylaspartate normalized to creatine. This ratio is lower in several brain regions in RIS compared to controls [82]. Conversely, a rise in serum levels of neurofilament-light (NFL) chain reflects neurodegeneration. NFL is released upon neuroaxonal injury into the blood stream, and its increase in blood occurs from RIS [83,84]. Thus, to slow the progression of MS pathology and disability, treatment to protect axons and neurons must be initiated very early. 

To slow the neuropathology of MS, an understanding of the mediators of injury is required. These mediators include secretory products of immune cells such as cytokines and proteases [6,85,86], mitochondrial injury [87], glutamate excitotoxicity [88,89] and oxidative stress. The presence of oxidative stress in MS lesions is well reviewed [90,91] and informed by the presence in lesions of lipid peroxides and their breakdown aldehydes such as malondialdehyde and 4-hydroxy-2-nonenal, and by oxidized DNA including 8-hydroxy-2′-deoxyguanosine. Van Horssen et al. [92] found that these markers were elevated in active demyelinating lesions, while Nikic et al. [93] associated reactive oxygen and nitrogen species with focal axonal degeneration. In general, oxidative stress is commonly observed in active MS lesions of both the gray and white matter [29]. This is compounded by the decline in anti-oxidant enzymes that detoxify free radicals [94], and by a decrease in the thiol glutathione [95,96] over the course of MS. There is a need for anti-oxidants that penetrate into the brain in MS, and this is the basis of clinical trials with alpha lipoic acid in MS [32]. Dimethylfumarate used in MS to control aberrant lymphocyte activity activates the transcriptional activator Nrf2 [97] that lies upstream of many anti-oxidant enzyme systems. Whether this contributes to its efficacy in MS is unknown.

Another mediator of injury that is increasingly appreciated in MS is iron. Its accumulation noted from early MS is detailed in the Introduction. The source of iron accumulation in MS brains is still unclear, but oligodendrocytes are known to be a major cell type that accumulate iron, since iron is an important cofactor for several enzymes in myelin synthesis [7,98,99]. In the human brain, iron is also stored in ferritin in myelin sheets [100]. Upon the destruction of oligodendrocytes and myelin, iron is released extracellularly and can be phagocytosed by microglia and macrophages [28]. If these myeloid cells degenerate, there can be an additional wave of iron deposition and associated oxidative stress [101]. The neurotoxicity of iron is likely exacerbated by the reduced level of the ferroptosis inhibitor, glutathione peroxidase-4, in the gray matter of MS brains [102]. The toxicity of ferrous iron involves oxidative stress, with the highly reactive hydroxyl radical formed from the Fenton reaction of ferrous iron with H2O2. However, iron is also destructive through other mechanisms, including ferroptosis [12,13], activating sphingolipid death pathways [16], and in misfolding proteins that compromise cells [17,18,19,20]. 

Despite the toxicity of iron, none of the disease-modifying therapies used in MS are known to affect iron homeostasis. Herein, we unexpectedly found that extracts of HS applied to human or mouse neurons in culture protected against iron neurotoxicity, and that HS attenuated axonal loss that accompanies the EAE disease. The protective capacity of HS against the iron-mediated killing of neurons is not directly related to anti-oxidant activity, as potent anti-oxidants (ferulic acid, and partially alpha lipoic acid) could not alleviate the killing of neurons by iron, and because the F1 fraction with low HORAC or ORAC activity ameliorated iron neurotoxicity. The activity of HS is not the direct result of its anthocyanin content, as the F1 fraction had a low level, and because a commercial preparation with high anthocyanin content was not protective. HS also protected against the killing of neurons by iron when several MS disease-modifying therapies could not. 

The mechanism(s) by which HS protects against the killing of cultured neurons by iron is unknown, and it is possible that HS mobilizes several defense mechanisms within cells. In support, HS protects neurons against diverse mediators of injury, including amyloid-beta, rotenone and staurosporine (Figure 4). Continuing studies in this laboratory aim to delineate how HS is neuroprotective against iron, and whether specific constituents or combinations within HS lead to its capacity to be profoundly protective against neuronal death. 

Our study has several drawbacks. First, MS is a chronic disorder, while iron exposure in tissue culture is acute, so the link between the profound killing of neurons by iron in vitro to the neurodegeneration that occurs over years in MS is tenuous. Second, the mechanisms by which iron kills neurons acutely in culture are not known, so the precise targets of HS in ameliorating iron neurotoxicity in culture are uncertain. Iron can kill cells by several mechanisms, as noted earlier, including through oxidative stress, ferroptosis and misfolding of proteins, and the extent to which HS neutralizes each of these remains to be determined. Another shortcoming is that we did not evaluate how iron destroys neurons in culture and how HS could have protected against these mechanisms. As mentioned above, it is plausible that HS mobilizes several defense mechanisms within cells that confer protection, as noted by the capacity of HS to protect not only against iron but also other stressors. However, this plausibility will need to be examined in future studies. 

## Figures and Tables

**Figure 1 cells-11-00440-f001:**
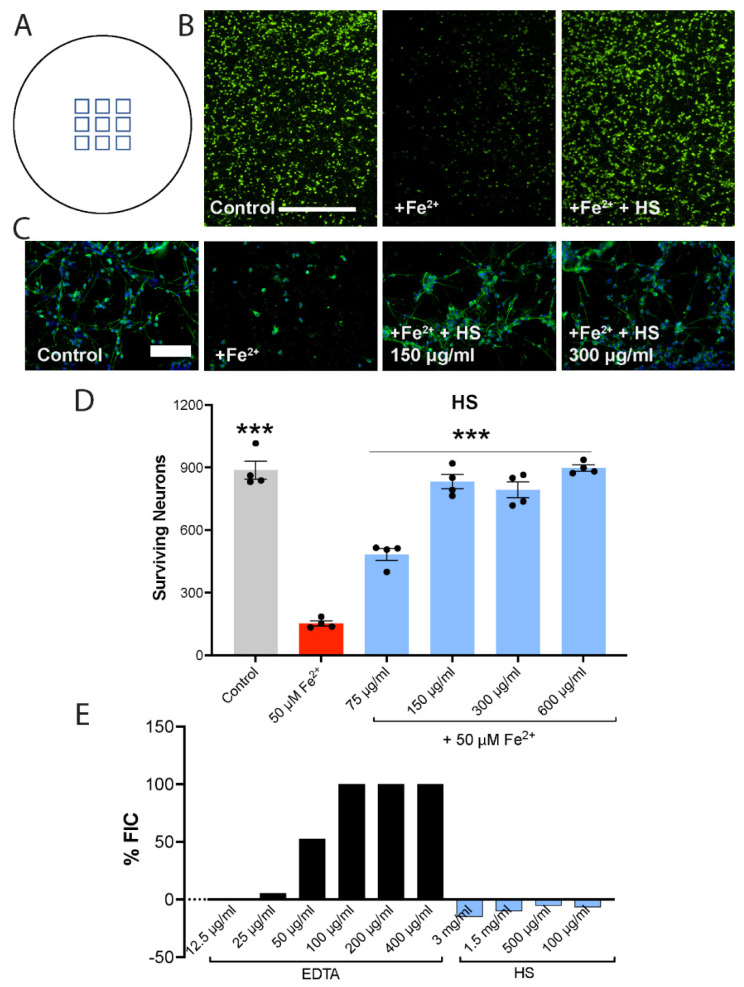
The killing of human neurons by ferrous iron is prevented by HS. (**A**) Schematic representation of the 9 (or 12 in some experiments) fields per view per well of a 96-well plate that were counted by the automated ImageXpress. The total count of neurons across these 9 fields is documented to represent the number of neurons that survived in that well. (**B**) A low-magnification image (captured using a 4× objective lens) each of a control, iron alone or iron + HS was captured well so as to visualize the substantial loss of neurons in response to iron and the protection by HS. Scale bar is 100 μm. (**C**) Tuj1 staining (green) to label tubulin β3 of neurons shows iron (Fe^2+^, 50 μM)-mediated loss of human neurons over 24 h, which is prevented by the presence of HS. Blue: DAPI to label nuclei. The scale bar in the control group is identical across all 4 panels and represents 100 μm. (**D**) In this experiment, the killing of human neurons by Fe^2+^ is reduced partially by 75 μg/mL HS and completely by higher HS concentrations. Values are mean ± SEM, with each dot representing a well. The protection by HS of iron neurotoxicity is reproduced in two other human preparations. *** *p* < 0.001 compared to iron (1-way ANOVA with Dunnett multiple comparisons). (**E**) The *y*-axis shows ferrous iron-chelating (FIC) ability. While EDTA chelates iron (upward direction of histogram) effectively, HS does not, even at very high concentration of 3 mg/mL. We do not know the reason for the negative values, but it appears that high concentrations of HS interfered with the assay. The lack of chelation of iron by HS is replicated in another experiment.

**Figure 2 cells-11-00440-f002:**
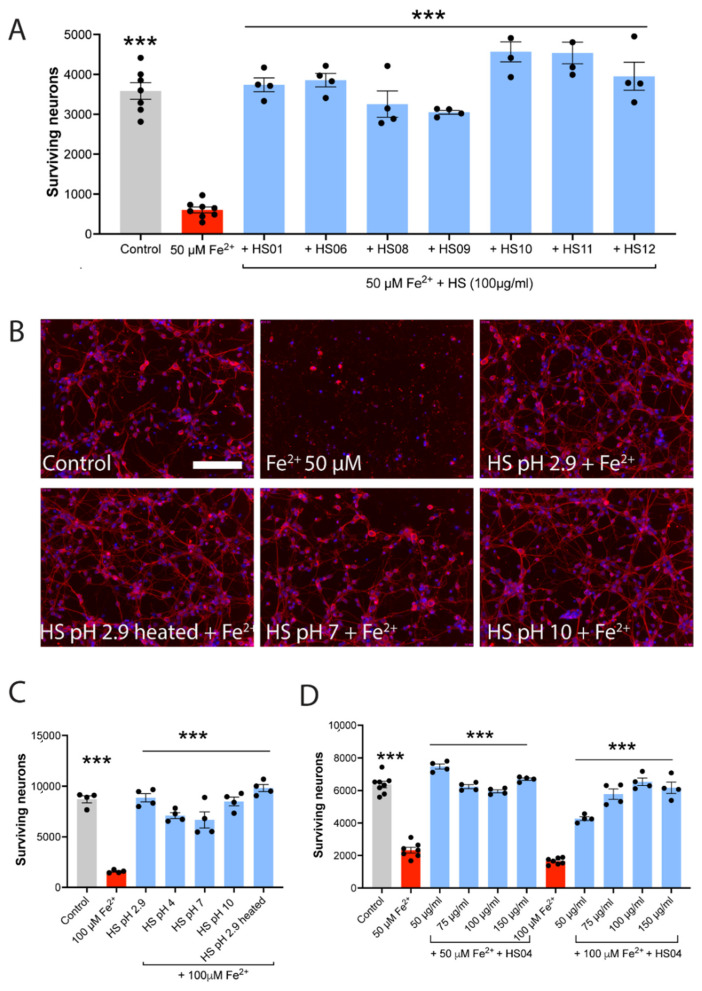
HS of varied pH protects against the iron-mediated killing of mouse neurons. (**A**) The killing of mouse neurons by iron is completely prevented by 7 different batches of HS extracts. All groups are different from iron alone: *** *p* < 0.001 (1-way ANOVA with Dunnett multiple comparisons). (**B**,**C**) The protection of mouse neurons against iron toxicity by HS occurs despite the pH of HS being altered or HS being heated. Neurons are detected by Tuj1 staining (red), and all cells are labeled with DAPI (blue). The scale bar in the control group is identical across all 6 panels and represents 100 µm. Values are mean ± SEM, with each dot representing a well. (**D**) Complete protection against iron neurotoxicity occurs from 50 to 150 µg/mL HS when the iron concentration is 50 µM. When FeSO_4_ is raised to 100 µM, the 50 µg/mL HS is less but still highly effective in ameliorating neuronal death. *** *p* < 0.001 (1-way ANOVA with Dunnett multiple comparisons) compared to iron.

**Figure 3 cells-11-00440-f003:**
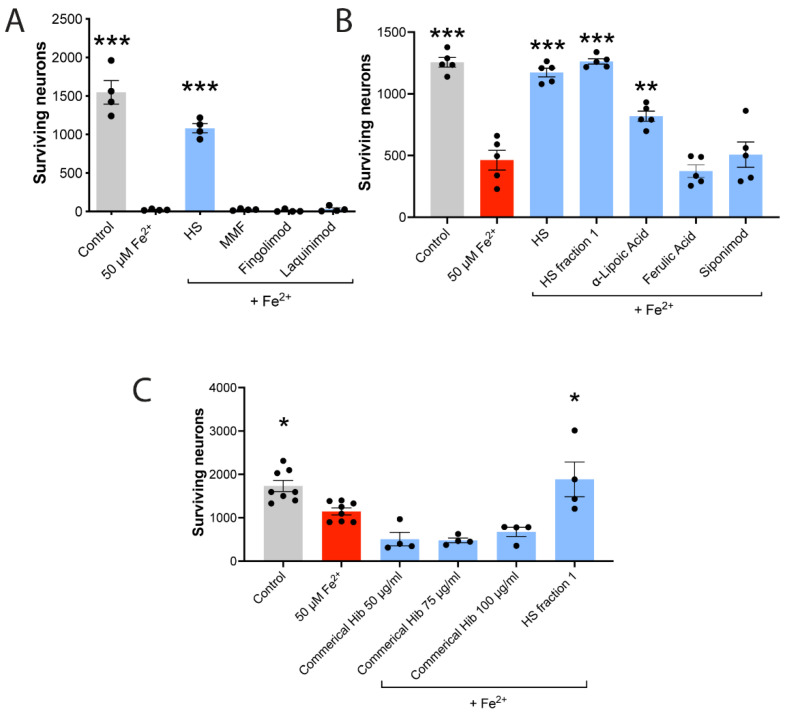
HS protection against iron neurotoxicity is not mimicked by MS medications or a commercial hibiscus extract and is not proportional to anti-oxidant activity. (**A**) The loss of neurons caused by 50 µM of FeSO_4_ is ameliorated by HS (100 µg/mL) but not by MMF (100 µM), fingolimod (FTY-720P, 100 nM) or laquinimod (5 µM). These concentrations of drugs used in MS are congruent with reported concentrations for neuroprotection against other stressors in the literature. (**B**) HS (100 µg/mL) and its fraction 1 (80 µg/mL) completely overcome iron toxicity, while the strong anti-oxidant ferulic acid (5 µg/mL) or the MS medication siponimod (1 nM) are ineffective. *** *p* < 0.001 (1-way ANOVA with Dunnett multiple comparisons) compared to FeSO_4_. Alpha lipoic acid (50 µg/mL) partially ameliorates killing of neurons by iron. (**C**) The commercial hibiscus from 50 to 100 µg/mL does not protect iron neurotoxicity unlike the positive control, HS fraction 1. All values are mean ± SEM, with each dot representing a well. * *p* < 0.001, ** *p* < 0.01, *** *p* < 0.001 (1-way ANOVA with Dunnett multiple comparisons) compared to iron.

**Figure 4 cells-11-00440-f004:**
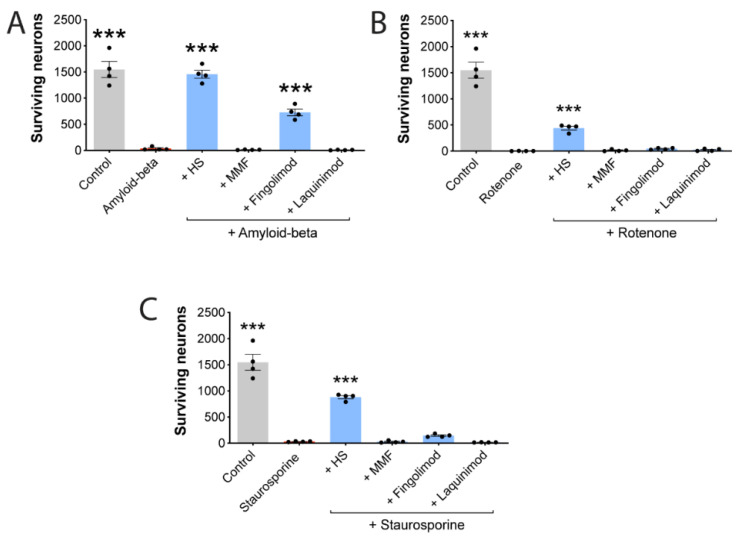
HS protects against other mediators of injury. Neurons were exposed to HS and the test medications for 1 h prior to amyloid-beta (20 µM) (**A**), rotenone (10 µM) (**B**) or staurosporine (200 nM) (**C**). Cells were harvested 24 h after for neuronal counts. HS was used at 100 µg/mL, while MMF was tested at 100 µM, fingolimod (FTY-720P) at 100 nM and laquinimod at 5 µM. All values are mean ± SEM, with each dot representing a well. *** *p* < 0.001 compared to the toxins alone (1-way ANOVA with Dunnett multiple comparisons).

**Figure 5 cells-11-00440-f005:**
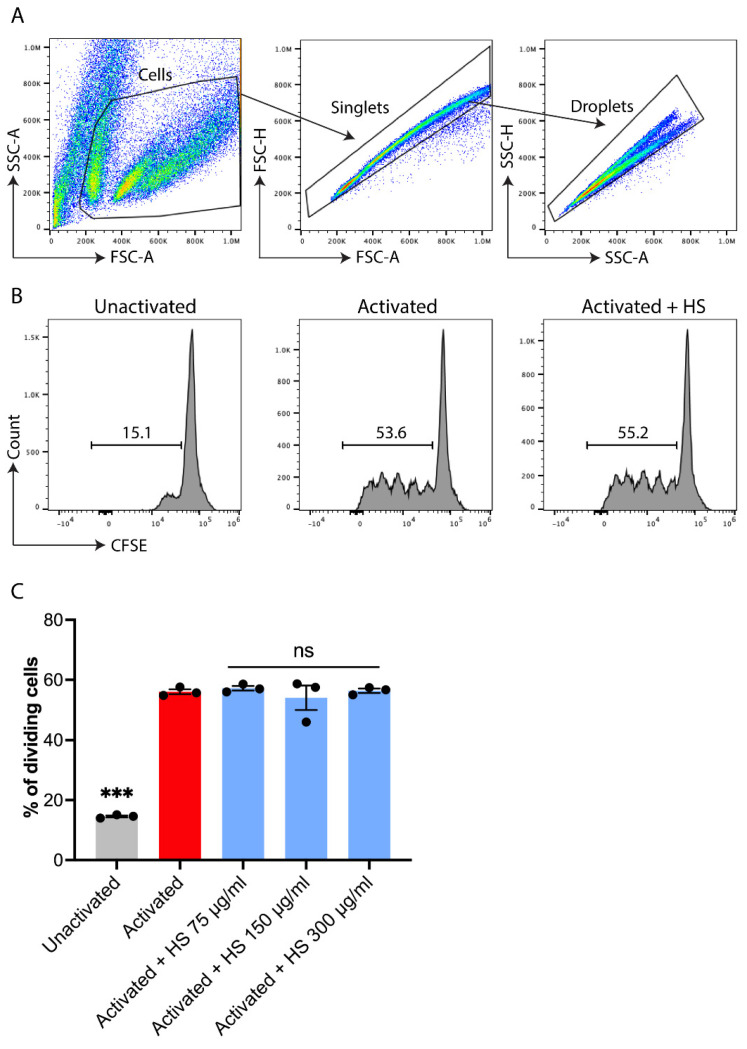
HS does not reduce the proliferation of T cells in culture. (**A**) FACS plots showing how the cells are evaluated. (**B**) CFSE plots where the humps to the left of the tall peak represent cycles of cell division. The % of dividing cells is also displayed. (**C**) Mean ± SEM of triplicate experiments. *** *p* < 0.001 (1-way ANOVA with Dunnett multiple comparisons) compared to activated controls. ns: not significant different from activated controls.

**Figure 6 cells-11-00440-f006:**
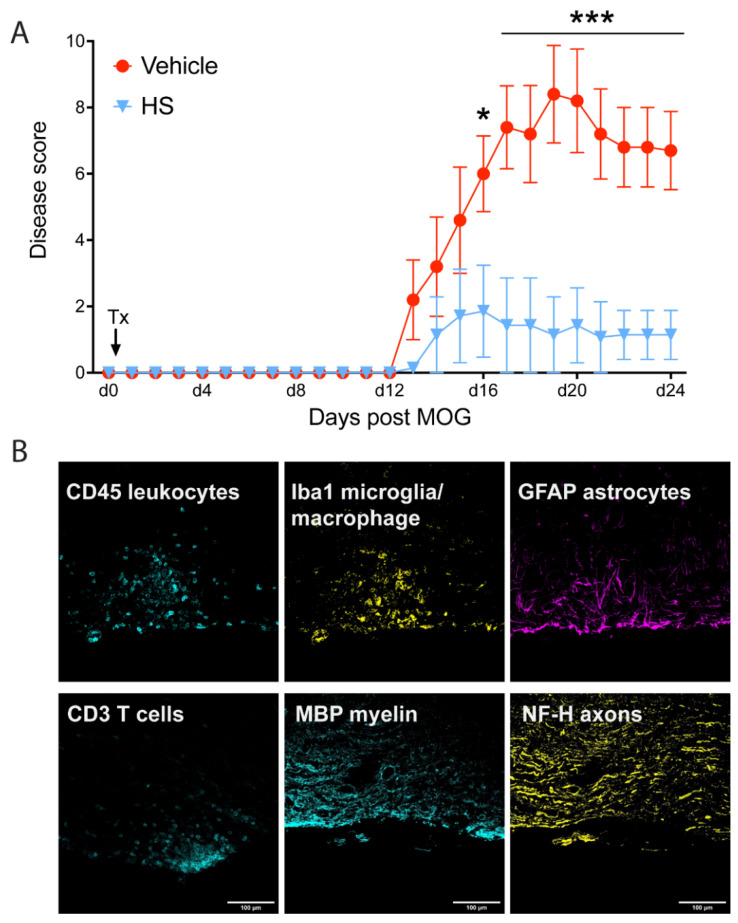
Oral HS treatment of mice immunized for EAE attenuates clinical disability. (**A**) While all 7 mice treated with vehicle develop severe EAE, this is significantly attenuated when mice are treated with HS (n of 5 mice). * *p* < 0.05, *** *p* < 0.001 (two-way ANOVA with repeated measures). Treatment initiation of the daily oral gavage is indicated by ‘Tx’. Please see Appendix A for the disability in mice representative of both groups. (**B**) Longitudinal thoracic sections of the spinal cord are stained for the various markers, and here, high-magnification images are displayed of a lesion (hypercellularity) of leukocytes (CD45+), some of which are CD3+ T cells and Iba1+ macrophages, along with microglia/macrophages (Iba1+), and reactive astrocytes (GFAP+); the respective colors are defined by the words in each panel. The edge of the tissue is at the bottom part of each panel. Disruption of MBP+ (myelin basic protein+) myelin and NFH+ (neurofilament heavy chain) axons can be seen in that lesion. Scale bar is 100 µm.

**Figure 7 cells-11-00440-f007:**
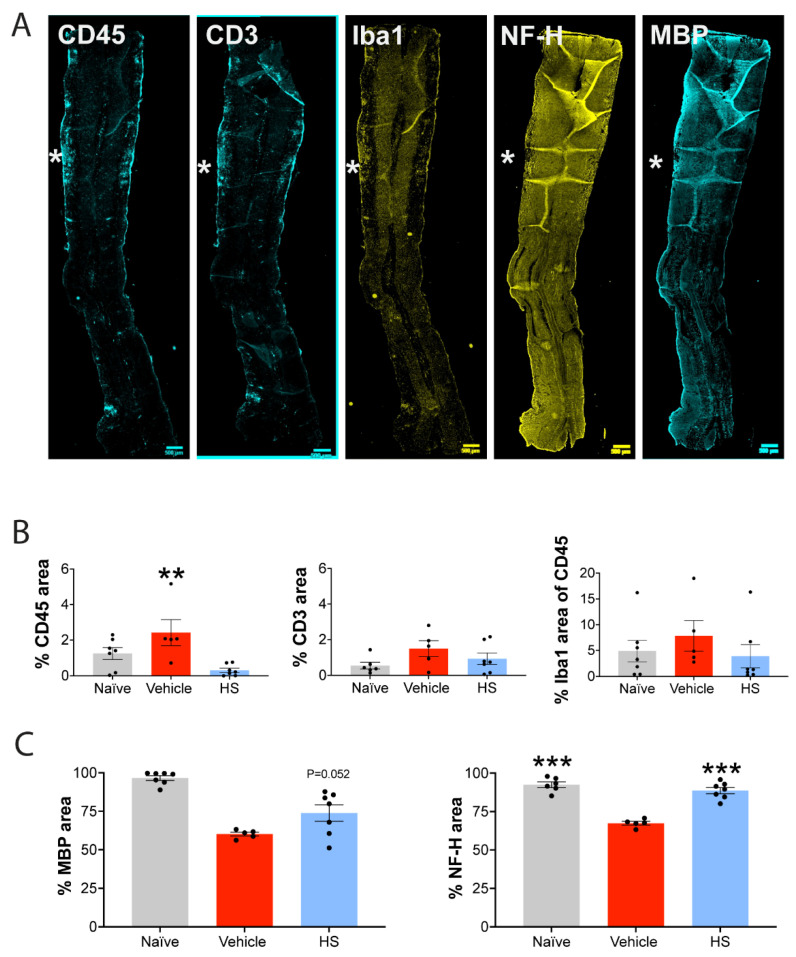
HS treatment prevents axonal loss in EAE. (**A**) Longitudinal thoracic spinal cord sections displayed with an active lesion indicated by *. Scale bar is 500 µm. These longitudinal sections were used for analysis of the spinal cord covered by a particular stain. (**B**) Quantitation of the extent of neuroinflammation (CD45 for immune cells; CD3 for T cells; Iba1 for microglia/macrophage) in the spinal cords of control-EAE or HS-treated EAE mice, killed at day 24 (from the experiment of Figure 6). ** *p* < 0.01 (2-tail unpaired *t*-test). (**C**) Quantitation of the extent of myelin (MBP) and axonal (NFH) preservation in the spinal cord of EAE mice treated with HS. Naïve mice referred to as spinal cords from unmanipulated animals. *** *p* < 0.001 compared to EAE-vehicle control (1-way ANOVA with Dunnett multiple comparisons).

**Table 1 cells-11-00440-t001:** The HORAC and ORAC activities do not predict protection against neuronal killing.

Sample	HORAC (μmole/g)	ORAC (μmole/g)	Relative Protection against Iron Killing (%)
HS	575 ± 75 (10 extracts)	480 ± 45 (3 extracts)	100
Commercial hibiscus	>800	Not determined	0
HS Fraction 1	0	0	100
Alpha lipoic acid	848	300	~50
Ferulic acid	4549	7451	0

## Data Availability

Available upon request.

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
