# Peer review of "A Distinct Hibiscus sabdariffa Extract Prevents Iron Neurotoxicity, a Driver of Multiple Sclerosis Pathology"

_cells, 2022, doi:10.3390/cells11030440_

Round 1
Reviewer 1 Report
Dear Author, Thanks for submitting your research manuscript entitled “A distinct hibiscus sabdariffa extract prevents iron neurotoxicity, a driver of multiple sclerosis pathology”
Before the final comments on this manuscript, the author needs to address the following comments in a scientific manner.
Major concerns:-
Please find out the following comments
The reviewer find the paper so interesting, as the author had performed substantial parameters. However there are some points which are highly imposing like comparing Hibiscus sabdariffa with other clinical drugs used in Multiple sclerosis.
The neurotransmitters assessment is missing from whole manuscript which is the major drawback.
Abstract:
The Abstract is written in a scientific way.
Introduction:
The author had described the introduction in a well-defined manner, but it would be more informative if the author had mentioned the symptoms of multiple sclerosis along with clinical reference.
Material and methods:
- In EAE & Spinal cord histology
The author should have mentioned the weight of C57BL/6 female mice.
- Behaviour parameters
The author should have performed other behaviour parameters to evaluate the toxicity caused by iron in Multiple sclerosis
- Morris water maze
- Beam crossing task
- Rota Rod test
- The Author need to estimate the level of neurotransmitter as iron toxicity is associated with disturbance in the neurotransmitter.
Author Response
In response to Reviewer 1:
“The reviewer find the paper so interesting, as the author had performed substantial parameters. However there are some points which are highly imposing like comparing Hibiscus sabdariffa with other clinical drugs used in Multiple sclerosis. The neurotransmitters assessment is missing from whole manuscript which is the major
drawback.”
We thank the Reviewer for the kind comment that the paper is interesting, and that we had provided substantial parameters.
In Figure 3A, we compared Hibiscus sabdariffa (HS) to MS medications which did not protect against iron neurotoxicity. We have now added this important comparison data in the abstract. Thank you for this comment.
The Reviewer raises a valid comment that the assessment of neurotransmitters is missing from the manuscript. We agree that this is a major drawback in the EAE field in general. We have addressed this concern in the Reviewer’s last comment below that refers to this question.
“Abstract:
The Abstract is written in a scientific way.”
Thank you for this comment.
“Introduction:
The author had described the introduction in a well-defined manner, but it would be more informative if the author had mentioned the symptoms of multiple sclerosis along with clinical reference.”
Thank you. We have now included a sentence in the Introduction on the symptoms of MS (lines 31-34) and have cited a classic reference on the topic (new reference #1).
“Material and methods:
In EAE & Spinal cord histology. The author should have mentioned the weight of C57BL/6 female mice.”
We apologize for this oversight. We have now included this in the Methods describing EAE (line 217-218).
“Behaviour parameters
The author should have performed other behaviour parameters to evaluate the toxicity caused by iron in Multiple sclerosis
Morris water maze
Beam crossing task
Rota Rod test”
Thank you for this excellent comment. We did not perform these behavioral tests in our experiments as EAE disability scores have typically been evaluated using the clinical disability scores described in the enclosed work. In support, recent publications by others in the Nature series journals in 2021 use similar disability scores and not the Morris water maze, beam crossing or rotarod. Examples of these publications include: Khaw et al., Nature Comm 12:105, 2021; Shi et al., Nature Comm 12:2547, 2021; Grigg et al., Nature 600:707, 2021.
One reason why the Morris water maze, beam crossing or rotarod tests are not commonly used in EAE research is that disabled EAE mice simply fail to perform once they have reached a certain level of disability. Nonetheless, the Reviewer has raised an excellent point of behavioral impairment, so we now include a new Supplementary video 2 that demonstrates a representative mouse from HS or vehicle-treated EAE group so that the reader can clearly see the marked differences between the animals. This new data is now described in lines 454-459.
“The Author need to estimate the level of neurotransmitter as iron toxicity is associated with disturbance in the neurotransmitter.”
As noted above, the lack of measurement of neurotransmitter changes is a major drawback in the EAE field. However, the EAE field is uncertain in general as to which neurotransmitter system to assess, since the EAE animal model is unpredictable in which part of the spinal cord the histological changes will occur in an animal, and this varies across animals. For this reason, our group has provided axonal counts of the entire longitudinal extent of the thoracic spinal cord instead in Figure 7C, as this should account for the uncertain axonal tracts that would be affected in one animal to the next. We now clarify this in the Methods (lines 259-266) that:
“It would be of important to assess changes to neurotransmitter systems in EAE and to determine whether these are rescued by HS. However, EAE is unpredictable in the location of lesions in the spinal cord, and EAE neuropathology is affected differently from one mouse to the next, so that it would be difficult to assess accurately and reliably the changes to neurotransmitters in an EAE experiment. For this reason, we developed quantitative axonal counts through the white matter of the entire thoracic cord (54), so that changes to axonal perturbation from one mouse to another could be meaningfully captured across a large area of the spinal cord in an experiment.”
We thank the Reviewer for these insightful comments that have improved the quality of the manuscript.
Reviewer 2 Report
In the manuscript Cells-1536422 by Mishra et al, authors have used Hibiscus extract (HS) against iron induced toxicity in neurons in culture conditions. They further extended their efforts to examine the protective efficacy of HS in mice model of MS. Finally, they propose the use of HS for MS. Their finding is interesting and very unexpected.
I have some major concerns and several points to be addressed.
Major concerns--
- Only readout of iron toxicity shown is the % cell survived. In the images showing FeSO4 treated neurons, it looks like within 24 hr all the cells are vanished after iron treatment. It would be important to understand the cause of this acute exposure of FeSO4. Did authors examine the iron toxicity parameters like cell death or oxidative stress, ferroptosis?
- Most of the neurological diseases including multiple sclerosis are chronic and progressive in nature. So, using an acute exposure (24hr) of FeSO4 in neurons should not be directly linked to MS without proper characterization of adversities.
- What is the cause of reduced cell survival in FeSO4 exposed group in this study?
- What parameters were examined for iron toxicity assessment. Knowing the cause of death upon iron exposure would indicate the possible mechanism of amelioration by HS.
- Significant variation in number of dead mouse neurons against FeSO4 exposure. There are approximately 4 times more surviving cells in 50uM FeSO4 exposed group in Figure 2D as compared to Figure2A. Similarly in Figure 3. Panel A, B and C. All of them have dramatic difference in number of surviving cells. Each of these experimental panel is mentioned to be performed in at least three replicates. This creates ambiguity. Please explain.
- What are the constituents of the HS extract solution used in this study?
- Alpha lipoic acid is a strong antioxidant and being used in clinical trials. However, HS Fraction 1 did not seem to have antioxidant capacity. How does HS extract protect the neurons?
There are several points-
- Line 3. “HS neuroprotection” please rephrase.
- Line 18. “iron killing” Please rephrase.
- Line 90. “In high concentrations”. It would be helpful to mention some range of the concentrations as well.
- Line 94. “capacity” please change this with “efficacy”. “toxicity of neurons”. Please check for correctness. “toxicity in neurons”.
- Line 95. “EAE”. This has not been defined in the manuscript.
- Line 118. Were these neurons obtained from patient/diseased brain or normal?
- Line 207. Please follow the standard statistical annotations. “T” is small in “t-test”. “P” of p-value should be italic. “Compare between 2 sample” or two groups?
- Result section-
- Line 213. Please change the “iron killing” throughout the manuscript.
- Line 215-216. What was the basis of using 50uM FeSO4 and also same goes for the selecting 75-600ug/mL of HS?
- Figure 1A. Looks like after 24 hr, all the cells are vanished in Iron exposed group. How do authors explain the this? Any readout of cell death?
- Please provide the scale. In Figure 1A, Fe+2 exposed panel does not appear to be in the same scale. The two cells look quite bigger here.
- Figure 1B. Bar for Fe+2 exposed group. With the provided representative images in Panel A (for Fe+2 exposed group), even the number looks way more than that. It seems like there no cells or a better representative image should be provided.
- In Figure 1C. What do the negative value mean here?
- Legend section for Figure 1. Please mention what the green and blue colors are indicating. I am assuming BLUE is DAPI and green is Tuj1 staining.
- Line 234. “Supplement the human results”. Please rephrase.
- Figure 2B. Any evidence of cell death? Did authors examine any marker? Please provide the scale for the images.
- Were neurons incubated with Fe+2 and HS simultaneously or at different time. Pls state the exposure regimen.
- Line 252-254. What concentration of these HS solution was used?
- Line 257-258. There is nothing to compare this. Do authors imply that they tested at lower concentration too?
- Line 262. “afforded”. Please rephrase.
- Line 270-273. It would be helpful to include some information about these drugs in introduction. It is important to at least mention mechanism of protection by these drugs in few lines.
- Line 274-275. Not clear. Which incubation was at first FeSO4 or the HS/MS medications?
- I am assuming HS incubation was followed by FeSO4? Please revise the sentence for correctness.
- Line 275. “Twenty-four after”. Please revise for correctness.
- Line 280-282. Could it be possible that the FeSO4 concentration being used here is too high. Neurodegeneration is a progressive process rather than the acute adverse effect.
- Line 292. what was the difference between Commercial HS and HS Fraction 1? Other than the anthocyanin level.
- Line 302-304. Interesting. Related to the above point about the difference between Commercial HS and HS-fraction 1. It appears that HS with low anthocyanin is effective.
- Line 305. Did they measure oxidative stress in cells upon FeSO4 exposure?? It would be important to see if FeSO4 results Oxidative stress in exposed cells.
- These results show that the used chemical HS Fraction one has no antioxidant capacity.
- Results are not consistent with the conclusions. As HS (top in the table 1) shows significant Antioxidant activity and in Figure 3B, it shows almost rescue similar to the HS Fraction 1. So, one chemical with antioxidant activity rescue one without antioxidant activity rescues. This further makes it important to examine the oxidative stress case in FeSO4 exposed cells.
- Did authors examined the effect of HS alone in control mice at the same dose used (250mg/kg)?
- Line 350. HS used here was HS Fraction 1?
- Line 353-359. Not clear?? Pls explain.
- Figure 5B. Are these images from immunized mice fed on HS? There is nothing to compare with. Based on the text in line 358-359 and legend section Figure 5B, the result and conclusion are not clear. It would be helpful to have some explanation. Please revise.
- Figure 6A. Are these images representing the disease condition or the rescue with HS?
- Figure 6B. Do authors have data from the naive mice? As having that data along with the vehicle and HS group would be helpful to strengthen the conclusion. Similar to the MBP and NF-H in panel C.
- Line 468. “protective factor”. Please rephrase.
- Line 469. “by our particular method”. This sentence emphasizes that extract preparation method has supplied some properties. Author must discuss this more. Otherwise revise the sentence to tone it down.
- Please revise the manuscript thoroughly for the correctness of sentences.
Author Response
In response to Reviewer #2:
“In the manuscript Cells-1536422 by Mishra et al, authors have used Hibiscus extract (HS) against iron induced toxicity in neurons in culture conditions. They further extended their efforts to examine the protective efficacy of HS in mice model of MS. Finally, they propose the use of HS for MS. Their finding is interesting and very unexpected.”
Thank you for the comment that our findings are interesting and unexpected.
“I have some major concerns and several points to be addressed.
Major concerns--
- Only readout of iron toxicity shown is the % cell survived. In the images showing FeSO4 treated neurons, it looks like within 24 hr all the cells are vanished after iron treatment. It would be important to understand the cause of this acute exposure of FeSO4. Did authors examine the iron toxicity parameters like cell death or oxidative stress, ferroptosis?”
Thank you for this important point. We have not interrogated the mechanisms by which iron kills neurons in culture, as we have focused our work on how to stop this acute killing effects of iron with generic medications (e.g., as reported in our Faissner et al., Nature Comm 8:1990, 2017 paper), or with HS in the current study. We have started only recently to address the mechanisms of iron killing of neurons in culture, but datasets are too preliminary to be included in this paper. We hope the Reviewer will accept our position that the current paper is a descriptive one of iron killing, and of HS protection, while we endeavor in future work to elucidate the detailed mechanisms of both.
We now include in the new second last paragraph of the Discussion (lines 577-589) the drawbacks of this manuscript, including that the mechanisms by which iron kills neurons acutely in culture are not known so the precise targets of HS in ameliorating iron neurotoxicity in culture are uncertain. We further state in that paragraph that “Iron can kill cells by several mechanisms as noted earlier including through oxidative stress, ferroptosis and misfolding of proteins, and the extent to which HS neutralizes each of these remains to be determined. Another shortcoming is that we did not evaluate how iron destroys neurons in culture, and how HS could have protected against these mechanisms. As mentioned above, it is plausible that HS mobilizes several defense mechanisms within cells that confer protection, as noted by the capacity of HS to protect not only against iron but also other stressors. However, this plausibility will need to be examined in future studies.”
To enable the reader a further appreciation of the potent killing afforded by iron, we now provide a new Supplemental Video 1 which records in real time the uptake of SYTOX (a DNA binding dye that crosses the compromised plasma membrane of cells) and where all cells are labeled with the CMPTX cell tracker. The video shows the progressive increase in number of SYTOX-positive cells when imaged 8 to 20 hours after iron exposure, and the prevention of this by HS.
“2. Most of the neurological diseases including multiple sclerosis are chronic and progressive in nature. So, using an acute exposure (24hr) of FeSO4 in neurons should not be directly linked to MS without proper characterization of adversities.
Thank you for this cautionary note. In the second last paragraph of the Discussion, we now state that “Our study has several drawbacks. First, MS is a chronic disorder while the iron exposure in tissue culture is acute, so the link between the profound killing of neurons by iron in vitro to the neurodegeneration that occurs over years in MS is tenuous.”
Moreover, we have deleted our previous last sentence of the Discussion of “Thus, we propose HS as a potential therapeutic strategy not only for progressive MS with well established neuroaxonal loss, but also from the earliest period of radiological isolated syndrome and clinically isolated syndrome of MS when axonal and neuronal injury is already apparent.”
“3. What is the cause of reduced cell survival in FeSO4 exposed group in this study?” “4. What parameters were examined for iron toxicity assessment. Knowing the cause of death upon iron exposure would indicate the possible mechanism of amelioration by HS.”
These are valid and important questions. We hope the Reviewer will accept our response posed to Question 1 above, i.e. that we have not interrogated the mechanisms by which iron kills neurons in culture, as we have focused our work on approaches to stop this killing. We acknowledge that the current paper is a descriptive one of iron killing, and of HS protection, and we have now included a new paragraph in the Discussion (lines 579-581) that one drawback of this manuscript is that the mechanisms by which iron kills neurons are not known so the precise targets of HS in ameliorating iron neurotoxicity in culture are uncertain.
It is possible that HS mobilizes several defense mechanisms within cells, and we have now provided new data that HS protects not only against iron but also other stressors (new Fig. 4). These stressors are amyloid-beta, rotenone and staurosporine. They are introduced as a new section in Methods and Results.
“5. Significant variation in number of dead mouse neurons against FeSO4 exposure. There are approximately 4 times more surviving cells in 50uM FeSO4 exposed group in Figure 2D as compared to Figure2A. Similarly in Figure 3. Panel A, B and C. All of them have dramatic difference in number of surviving cells. Each of these experimental panel is mentioned to be performed in at least three replicates. This creates ambiguity. Please explain.”
This is a very astute observation which we should have explained in the first place. In the Methods (lines 152-163), we now clarify as follows: “It is important to note that there is variation across different batches of cultures, but that all test conditions are controlled within an experiment. These variations across batches of cells prepared at different times are contributed by the initial health of the isolated cells (some preparations are healthier than others despite the desire to be as consistent as possible), different individuals isolating the cells in the first place, the density of the cell suspension for plating, the susceptibility of neurons to iron-mediated killing that can vary across preparations for reasons that are unclear, and whether 9 or 12 fields of view were quantited in ImageXpress. Such variability across batches is reflected in the number of control neurons that differ across experiments (see for example the y-axis numbers in Figures 1 and 2). However, each experiment is internally controlled to have the same conditions of assessment. Moreover, each key result is evaluated over 3 different batches of neuronal cultures in separate experiments.”
“6. What are the constituents of the HS extract solution used in this study?”
This is an excellent question which we are hoping to resolve so that the active ingredients for neuroprotection could be isolated and used as a therapeutic. However, as a natural complex mixture, the constituents are still being worked through. In the Methods under “Preparation of HS”, we now state: “The composition of the complex aqueous extract is only partially characterized at this point, but it has low anthocyanin content compared to a commercially available capsule (see Results), several organic acids (e.g. glyceric acid, propionic acid, malic acid and oxalic acid) as determined by mass spectrometry, and several polyphenols (e.g. gallic acid, gallocatechin and kampferol) as evaluated by liquid chromatography (data not shown). The extraction method and content are being applied for intellectual protection.”
“7. Alpha lipoic acid is a strong antioxidant and being used in clinical trials. However, HS Fraction 1 did not seem to have antioxidant capacity. How does HS extract protect the neurons?
Indeed, this is an important question. We have cited references (12-20) that ferrous iron can destroy cells through several mechanisms, one of which is oxidative stress. For protection against iron-induced killing of neurons in our study, anti-oxidant capacity is not sufficient as summarized in Table 1. As discussed above in the rebuttal to Questions 3 and 4, it is possible that HS mobilizes several defense mechanisms within cells. In support, we have now provided new data that HS protects not only against iron but also amyloid-beta, rotenone and staurosporine (new Fig. 4). We hope the Reviewer accepts our statement that the precise mechanisms will have to be determined in future studies (lines588-589).
“There are several points-
- Line 3. “HS neuroprotection” please rephrase.
Thank you. This is now rephrased to “Neuroprotection by HS’
- Line 18. “iron killing” Please rephrase.”
This is now rephrased to “the killing of neurons by iron”
“3. Line 90. “In high concentrations”. It would be helpful to mention some range of the concentrations as well.”
We now have done so: “>50 mM of hibiscus delphinidin 3-sambubioside”.
“4. Line 94. “capacity” please change this with “efficacy”. “toxicity of neurons”. Please check for correctness. “toxicity in neurons”.”
Thank you. We have corrected both.
“5. Line 95. “EAE”. This has not been defined in the manuscript.”
We apologize for this omission and have now spelled out EAE at its first mention.
“6. Line 118. Were these neurons obtained from patient/diseased brain or normal?”
We do not know the nature of the brain specimens. We now state that “Ethics guidelines preclude access to patient information and the cause of the abortion.”
“7. Line 207. Please follow the standard statistical annotations. “T” is small in “t-test”. “P” of pvalue should be italic. “Compare between 2 sample” or two groups?”
Thank you for your thoroughness. We have corrected these, and also edited the legend to figures accordingly.
“8. Result section-
- Line 213. Please change the “iron killing” throughout the manuscript.”
Thank you, and we have now done so.
“10. Line 215-216. What was the basis of using 50uM FeSO4 and also same goes for the selecting 75-600ug/mL of HS?”
We now explain that 50 mM FeSO4 was used in a previous study, and that 75-600 mg/ml HS were concentrations that were empirically chosen to span a range,
“11. Figure 1A. Looks like after 24 hr, all the cells are vanished in Iron exposed group. How do authors explain the this? Any readout of cell death?
We now provide a new Supplementary Video 1 of live imaging of neurons in response to iron. The video shows that the number of neurons with compromised plasma membrane (SYTOX+) progressively increases following iron exposure. We believe that this loss of cellular integrity eventually results in cell detachment and thus loss of neurons.
“12. Please provide the scale. In Figure 1A, Fe+2 exposed panel does not appear to be in the same scale. The two cells look quite bigger here.”
Thank you for this comment. We have now replaced the iron-exposed picture with another image that is more representative of that group, as determined by the quantitative count in panel B.
“13. Figure 1B. Bar for Fe+2 exposed group. With the provided representative images in Panel A (for Fe+2 exposed group), even the number looks way more than that. It seems like there no cells or a better representative image should be provided.
Thank you for this critique. We have now replaced the iron-exposed picture with another image that is more representative of that group, as determined by the quantitative count in panel B.
“14. In Figure 1C. What do the negative value mean here?
We do not know the reason for the negative values, It is possible that high concentrations of HS interfered with the assay, which we now mention in the figure legend, but we have no proof for this.
“15. Legend section for Figure 1. Please mention what the green and blue colors are indicating. I am assuming BLUE is DAPI and green is Tuj1 staining.”
Our apologies. We have now identified the colors.
“16. Line 234. “Supplement the human results”. Please rephrase.”
We have changed ‘supplement’ to ‘corroborate’.
“17. Figure 2B. Any evidence of cell death? Did authors examine any marker? Please provide the scale for the images.”
We hope that Supplementary Video 1 of live imaging of neurons in response to iron now provides evidence of loss of cell integrity.
Scale is now provided.
“18. Were neurons incubated with Fe+2 and HS simultaneously or at different time. Pls state the exposure regimen.”
Sorry for this omission. In Methods (line 141), we now state that “In experiments involving test reagents (e.g. HS or medications used in MS) and iron, the test reagents were applied to neurons one hour before the iron.”
“19. Line 252-254. What concentration of these HS solution was used?”
We have now added ‘100 mg/ml’ HS in that paragraph.
“20. Line 257-258. There is nothing to compare this. Do authors imply that they tested at lower concentration too?”
Agree. We have removed that sentence.
“21. Line 262. “afforded”. Please rephrase.”
This has been changed to ‘achieved’
“22. Line 270-273. It would be helpful to include some information about these drugs in introduction. It is important to at least mention mechanism of protection by these drugs in few lines.”
We have now added that these drugs have neuroprotective effects through mechanisms that include anti-oxidant activity. In the Introduction, these were mentioned as ‘disease modifying therapies used in MS’.
“23. Line 274-275. Not clear. Which incubation was at first FeSO4 or the HS/MS medications?”
24. I am assuming HS incubation was followed by FeSO4? Please revise the sentence for correctness.”
We have clarified this to “Thus, human neurons were exposed first to HS, MMF, fingolimod or laquinimod, and this was followed 1h later by FeSO4”.
“25. Line 275. “Twenty-four after”. Please revise for correctness.”
We have now added in the missing ‘hours’.
“26. Line 280-282. Could it be possible that the FeSO4 concentration being used here is too high. Neurodegeneration is a progressive process rather than the acute adverse effect.”
This is a valid point. With regards to siponimod being ineffective against iron toxicity, we have now added to that sentence “in the conditions tested”.
“27. Line 292. what was the difference between Commercial HS and HS Fraction 1? Other than the anthocyanin level.”
We did not determine other potential differences between commercial HS and HS Fraction 1, so we have now noted in that paragraph: “Other differences in chemical constituents may differ between HS fraction 1 and commercial hibiscus but we did not document these.”
“28. Line 302-304. Interesting. Related to the above point about the difference between Commercial HS and HS-fraction 1. It appears that HS with low anthocyanin is effective.”
Yes, this speaks to the complexities of extracts from plants that contain not only anthocyanins but also other chemical constituents. We are trying to define the active species in HS that protect against iron neurotoxicity but it has been challenging.
“29. Line 305. Did they measure oxidative stress in cells upon FeSO4 exposure?? It would be important to see if FeSO4 results Oxidative stress in exposed cells.”
Unfortunately, we did not do this experiment. In line 584-585 of Discussion, we now state that “Another shortcoming is that we did not evaluate how iron destroys neurons in culture, and how HS could have protected against these mechanisms.”
“30. These results show that the used chemical HS Fraction one has no antioxidant capacity. Results are not consistent with the conclusions. As HS (top in the table 1) shows significant Antioxidant activity and in Figure 3B, it shows almost rescue similar to the HS Fraction 1. So, one chemical with antioxidant activity rescue one without antioxidant activity rescues. This further makes it important to examine the oxidative stress case in FeSO4 exposed cells.”
We accept these points and have thus introduced a new paragraph (second last one) in Discussion of the shortcomings of our study. That one chemical with antioxidant activity rescues while another antioxidant does not suggest that we could be dealing with several mechanisms of injury in our model. It will take time to resolve how iron is mediating killing in the neuronal cultures, and how HS could be protective. For now, I hope the Reviewer accepts our hope that this manuscript can be published as a descriptive one of the protection afforded by HS, while we continue to address the detailed mechanisms of injury and protection.
“32. Did authors examined the effect of HS alone in control mice at the same dose used (250mg/kg)?”
We did not examine HS alone in control mice, as pilot studies had informed us that there were no obvious behavioral changes caused by the plant extract by itself. We thus focused the experiment on whether HS could affect EAE severity.
“33. Line 350. HS used here was HS Fraction 1?”
We now clarify that this is ‘whole HS extract’ as used in Figure 1.
“34. Line 353-359. Not clear?? Pls explain.”
We now realize that the previous statements were confusing. We have removed them, so that the flow of the data transitioned from clinical disability of mice to histology without intervening commentary.
We have also introduced a new Supplementary Video 2 (lines 454-459) that represents the clinical deficits of EAE mice treated with vehicle or HS just before sacrifice at day 24 for histology.
“35. Figure 5B. Are these images from immunized mice fed on HS? There is nothing to compare with. Based on the text in line 358-359 and legend section Figure 5B, the result and conclusion are not clear. It would be helpful to have some explanation. Please revise.”
We apologize for the lack of clarity. The images of Figure 6B (previously 5B) are from an EAE animal treated with vehicle. We present Figure 6B as examples of stains that can be obtained in longitudinal sections of the spinal cord of EAE mice. We now describe (lines 462-463) that an active lesion in EAE is informed by an aggregate of CD45+ leukocytes, some of which are CD3+ T cells and Iba1+ microglia/macrophages.
“36. Figure 6A. Are these images representing the disease condition or the rescue with HS?”
We now state (lines 464-472), when introducing Figure 7A (previously 6A) that “As CNS lesions in EAE can vary in size and location, we quantified the histological outcomes by evaluating the thoracic spinal cord in longitudinal sections for each mouse. Figure 7A are examples of longitudinal sections used for quantitation. The images are from a vehicle-treated EAE spinal cord stained for CD45 (immune cells), CD3 (T cells) or Iba1 (microglia/macrophages), or for degree of myelin (myelin basic protein, MBP+) or axonal (neurofilament heavy chain, NFH) loss. Sections with a central canal were chosen for quantification to normalize location across mice. For quantitation, slide scanner images of both the right and left lateral columns were processed by ImageJ software for the total area per section bordered by the stain of interest, as described recently (54).”
“37. Figure 6B. Do authors have data from the naive mice? As having that data along with the vehicle and HS group would be helpful to strengthen the conclusion. Similar to the MBP and NF-H in panel C.”
Thank you for this. We have now included the naïve data.
“38. Line 468. “protective factor”. Please rephrase.”
We have changed ‘protective factor’ to ‘therapy’
“39. Line 469. “by our particular method”. This sentence emphasizes that extract preparation method has supplied some properties. Author must discuss this more. Otherwise revise the sentence to tone it down.”
Thank you. We have removed that last sentence entirely.
“40. Please revise the manuscript thoroughly for the correctness of sentences.”
We appreciate the Reviewer for having taken the time to evaluate the manuscript in a critical and thorough manner. We have revised the manuscript accordingly and believe it is now ready for publication. We hope the Reviewer agrees. Thank you again.
Round 2
Reviewer 2 Report
Authors have responded the major comments through text modification in which they accepted that the study has drawbacks/shortcoming and added in the manuscript text. Although, authors expressed that they are continuing to do future studies, an important concern is the confidence over the main and only readout of the in vitro part of the study i.e. cell survival number. This needs to be addressed. There are other points as well. Please my comments below.
- Line 14. “produced”??. Please rephrase it.
- Line 80. “However ameliorating ROS……”. Please tone down the sentence. Is it based on the experiments/studies? It more sounds like a forecast.
- Line 90. “have reported antioxidant”. Organic HS does have antioxidant properties. Do authors imply that during sub-fractionation process HS extract lost these properties?
- Line 93. Authors should include some lines about how HS is shown beneficial in these studies?
- Line 115. what did author mean by “intellectual protection”?
- Three different forms are used in this study. HS, HS fraction 1 and Commercial HS.
- Only HS fraction 1 did not show antioxidant activity and was used in Figure 3B.
- In the rest of manuscript, HS is used. Notable that it showed antioxidant activity (table 1).
- The creates confusion and raise questions!
- Line 133 and 140. it is notable that almost 75,000-100,000 cells were seeded in both human and mouse in vitro experiments, still the control group have ~900 to 6,000 cells. Line 147. Selection of these 9-12 view field may bring biased outcome. Line 152-163. This brings the major issue in the study. The only readout of their study is cell survival number as they have not extended their efforts to examine other toxicity readout. Moreover, there are variations in these number in different experiments. In the end of this paragraph, authors say that each result is based on over 3 different batches of neuronal culture in separate experiments. In each of their graphs (even for control) batches do have some variation (which is totally understandable). But this variation is huge when compared to the panels in different figures. How come different batches used in same figure panel show comparable numbers but then dramatically different from the other panel? Given that they mentioned in method section that a certain number of cells (75-100K cells) were seeded, it does cast doubt when different graphs have different cell survival numbers. In Figure 1B. ~900 out of 75000 cells? That means a significant area of the culture well had no cells surviving cells even in control group. In this case, selecting 9-12 view fields can be easily bring biased outcome. Why did author not used MTT assay to check cell survival? As it is a reliable and widely used assay for neuronal culture?
- Line 341. Result section “HS is superior to MS medications for neuroprotection against iron”. Are these MS medication act on neurons directly? or else they mainly act on immune cells? If they act on immune cells, why do authors compare the effect of these drugs with HS on neurons? Please remove this section.
- Line 365. Siponimod. This drug act on immune cells to minimize the MS symptoms. What is the purpose to use them on neurons?
- Line 571. “HS mobilizes several defense mechanisms “. What could be those several mechanisms? This is a wage sentence.
- Line 577. These are indeed drawbacks/shortcomings and needed to be addressed. At least for some of these to gain confidence on the study outcome.
Line 590. what is the purpose of this paragraph? This is just citing studies and ended with a shortcoming sentence.
Author Response
In response to Reviewer #2:
We thank the Reviewer for the prompt review of our revised manuscript, and for the feedback to improve our paper.
“Authors have responded the major comments through text modification in which they accepted that the study has drawbacks/shortcoming and added in the manuscript text. Although, authors expressed that they are continuing to do future studies, an important concern is the confidence over the main and only readout of the in vitro part of the study i.e. cell survival number. This needs to be addressed. There are other points as well. Please my comments below.”
Thank you for this important comment. We apologize for our oversight as we have not clarified sufficiently that the method of counting stained neurons in the current paper using automated ImageXpress is the most sensitive and reliable method that we have used over the last decade to document injury to neurons. We utilized this method to quantify cell death of neurons caused by activated microglia (Mishra et al., Annals Clin Trans Neurol 1:409, 2014), activated T cells (Sloka et al., PLOS One 10:e0114084, 2015), iron (Faissner et al., Nature Comm 8:1990, 2017; Faissner et al., Multiple Sclerosis J 24:1543, 2018; and Brown et al., Neurotherapeutics 18:387, 2021) and a novel neurotoxin that we recently identified, oxidized phosphatidylcholines (Dong et al., Nature Neurosci 24:489, 2021). We detailed this method of counting neurons (in response to oxidized phosphatidylcholines) in a step-by-step methods paper that we were invited by STAR Protocols (of CellPress) to write (Dong et al., STAR Protoc 2:100853, 2021), including ‘capture 9–12 imaging fields evenly distributed across each well per experimental condition to quantify neuron survival’.
Thus, our method of quantifying neuronal death has been in the published literature for some time. It is the best method that we have. Indeed, we have transitioned to the automated ImageXpress method from readouts of MTT (e.g. Giuliani et al., J Immunology 171:368, 2003), which the Reviewer kindly advised us as noted below, and ATP luminescence (Mishra et al., Annals Clin Trans Neurol 1:409, 2014). These spectrophotometric and fluorometric techniques were relatively insensitive in our hands, and did not provide the capability of ImageXpress where neurons that were counted could be visually inspected for corroboration (e.g new Fig. 1B). The automated ImageXpress where the machine evaluates a specified number of fields across all wells is an advance from the laborious manual counting of neurons that we used to do (e.g. Xue at al., Brain 132:26, 2009; and our publications prior to that), and where others are still doing if they do not have the resources to acquire an automated ImageXpress.
In the revised manuscript, we have now added (lines 158-161) that “Note that this ImageXpress method to quantify the loss of stained neurons, and potential protection by test compounds, has been used in our previous reports where neurons were destroyed by activated microglia (50), activated T cells (51), oxidized phosphatidylcholines (49) and iron (21, 52).”
We refer the Reviewer to Supplementary Video 1 which records in real time the uptake of a DNA binding dye that crosses the compromised plasma membrane of cells. The video illustrates the progressive increase in number of SYTOX-positive cells when imaged 8 to 20 hours after iron exposure, and the prevention of this by HS. Here, it should be clear that cell compromise is seen only in the iron group.
In Point #10 below, the Reviewer is concerned that neurons might not be reliably spread across a given well, so that preselecting a given number of fields of view may bias towards a particular result. This is a valid comment. We have now improved upon this statement (lines 153-155): “For each well of cells, 9 or 12 (identical within an experiment) fields of view (Fig. 1A) that are in the same locations in each well of every well of an experiment were imaged for quantitative analysis”. Moreover, we provide a new schematic (Fig. 1A) to clarify the fields of view within each well where neurons that had survived were counted. We also provide a new Figure 1B of neurons at low magnification, where the many cells visualized should give confidence that neurons are evenly spread out in control or HS-protected conditions, but where neurons are depleted in numbers in the iron-alone condition. The legend to Figure 1 has thus been accordingly edited.
Altogether, with the above clarifications and revisions, we hope that the Reviewer will now permit our work for publication. We thank the Reviewer for scrutinizing our methods, but hope that the Reviewer accepts our experience that the ImageXpress quantitation of neurons is superior to MTT or ATP luminescence. We hope that the Reviewer is reassured by our new paragraph (lines 158-161) that we have used the ImageXpress method for several neurotoxic (and potentially protective factors) factors in many journals (e.g. Nature Communications and Nature Neuroscience) over a decade.
“1. Line 14. “produced”??. Please rephrase it.”
Thank you. We have changed ‘produced’ to ‘collected’.
“2. Line 80. “However ameliorating ROS……”. Please tone down the sentence. Is it based on the experiments/studies? It more sounds like a forecast.”
We have now changed (lines 81,82) the sentence to: “However, ameliorating ROS produced by iron may not overcome iron toxicity, if other mechanisms noted above of iron-induced injury are prominent”.
“3. Line 90. “have reported antioxidant”. Organic HS does have antioxidant properties. Do authors imply that during sub-fractionation process HS extract lost these properties?”
We have removed the word “reported”.
“4. Line 93. Authors should include some lines about how HS is shown beneficial in these
studies?”
Thank you. In lines 93,94, we have clarified the benefit of HS in reducing brain pathology and biochemical indices of oxidative stress when injected intraperitoneally into mice in these models.
“5. Line 115. what did author mean by “intellectual protection”?
Thank you. We missed out ‘property’ in “The HS compositions are being applied for intellectual property protection”.
“6. Three different forms are used in this study. HS, HS fraction 1 and Commercial HS.”
“7. Only HS fraction 1 did not show antioxidant activity and was used in Figure 3B.”
“8. In the rest of manuscript, HS is used. Notable that it showed antioxidant activity (table 1).
- The creates confusion and raise questions!”
These questions refer to the same points so we are addressing them collectively. We apologize for our lack of clarity and for creating confusion. In the methods (lines 128-133), we now state: “In summary, in this study, we have tested HS as the aqueous extract of the dried calyxes of hibiscus sabdariffa plants, different HS batches to evaluate the reproducibility of extracts to protect against iron-mediated killing of neurons, a HS fraction 1 to determine whether a particular composition of HS retains its neuroprotective activity, and a commercially available hibiscus sabdariffa capsule to investigate whether it could replicate the neuroprotective activity of our HS extracts.”
“10. Line 133 and 140. it is notable that almost 75,000-100,000 cells were seeded in both human and mouse in vitro experiments, still the control group have ~900 to 6,000 cells. Line 147. Selection of these 9-12 view field may bring biased outcome. Line 152-163. This brings the major issue in the study. The only readout of their study is cell survival number as they have not extended their efforts to examine other toxicity readout. Moreover, there are variations in these number in different experiments. In the end of this paragraph, authors say that each result is based on over 3 different batches of neuronal culture in separate experiments. In each of their graphs (even for control) batches do have some variation (which is totally understandable). But this variation is huge when compared to the panels in different figures. How come different batches used in same figure panel show comparable numbers but then dramatically different from the other panel? Given that they mentioned in method section that a certain number of cells (75-100K cells) were seeded, it does cast doubt when different graphs have different cell survival numbers. In Figure 1B. ~900 out of 75000 cells? That means a significant area of the culture well had no cells surviving cells even in control group. In this case, selecting 9-12 view fields can be easily bring biased outcome. Why did author not used MTT assay to check cell survival? As it is a reliable and widely used assay for
neuronal culture?”
Thank you for these comments and we apologize for our previous lack of clarity. We hope our opening remarks above have clarified these points. We also hope that the new Figures 1A and 1B, which show that 9 or 12 fields in similar locations per well across all wells were counted, and that control cultures had cells that were evenly distributed across a well, have removed any uncertainties. Also, our method of quantitation has been reported in several papers from this lab the past 10 years, as noted in the citations above. Hopefully, with Figure 1A showing the fields of view in relation to the whole well, this clarifies how we have cells in the hundreds or thousands out of 75,000 cells seeded.
We did not use the MTT assay (which we have used in our earlier papers to document cell proliferation or death) as it is insensitive in our experience compared to the counts of neuronal cell numbers. MTT also reflects mitochondrial metabolism which may not always corresponds to cell toxicity. A stimulus that increases cell proliferation will elevate MTT values. The limitations of the MTT assay have been commented by others (e.g. van Tonder et al., BMC Research Notes 8:47, 2015).
With regards to our previous statement that “each key result is evaluated over 3 different batches of neuronal cultures in separate experiments”, we have now added “the trend of” at the beginning of the statement (line 172). This hopefully will provide confidence to our data that although the precise cell numbers may differ from one experiment to the next, iron kills neurons and this is protected by HS in each experiment.
“11. Line 341. Result section “HS is superior to MS medications for neuroprotection against iron”. Are these MS medication act on neurons directly? or else they mainly act on immune cells? If they act on immune cells, why do authors compare the effect of these drugs with HS on neurons? Please remove this section.”
Thank you for these comments. We wish to appeal this request to remove the section on “HS is superior to MS medications for neuroprotection against iron” as this is one of key points of the paper. While MS is an immune-mediated condition, substantial neurodegeneration occurs in the form of loss of neurons and axons. Thus, it is important to have neuroprotective medications, including against iron neurotoxicity which is not countered currently by any medications in MS. While MS medications are predominantly used to counter immune dysfunction in MS, they have been evaluated to determine if they would have incidental neuroprotective activity which would be advantageous for that drug.
To explain to the reader better, we have now included a new paragraph (lines 352-360) that:
“Medications used in MS are based on their capacity to reduce the dysregulated immune system in MS (58). However, with the recognition that MS has substantial degeneration of neurons and axons that drives the progression of disability (6, 8), there has been interest in whether MS medications have direct capacity to protect neurons. In this regard, dimethylfumarate [and also its metabolite monomethylfumarate (MMF) to which dimethylfumarate is rapidly converted in vivo] has been shown in culture to protect against H2O2- or amyloid-beta- induced death of neurons through an Nrf2-dependent pathway (59, 60), while fingolimod protects neurons against oxidative stress or glutamate excitotoxicity through activating sphingosine-1-phosphate receptors (61, 62). Another drug, laquinimod, reduces brain atrophy in patients with MS (63, 64), and has protective effects on neurons against various insults (50, 65), although it has not received regulatory approval for use in MS. Whether these could protect against iron neurotoxicity has not been addressed. Thus, we compared the potency of HS against these MS medications.”
“12. Line 365. Siponimod. This drug act on immune cells to minimize the MS symptoms. What is the purpose to use them on neurons?”
Hopefully, the new paragraph introduced in response to Reviewer’s point #11 clarifies that it is important not only to have an immune modulating agent, but also one with neuroprotective actions. We now describe (lines 372-376) that “In tissue culture, siponimod protects neurons against astrocyte-induced toxicity (66). The intraventricular infusion of siponimod in the EAE model of MS prevents synaptic degeneration and preserves neurons (67). However, when tested against iron in culture, siponimod was ineffective in conditions in which HS was completely protective (Fig. 3B).”
“13. Line 571. “HS mobilizes several defense mechanisms “. What could be those several
mechanisms? This is a wage sentence.”
This is a fair comment. Accordingly, we have removed that entire paragraph.
“14. Line 577. These are indeed drawbacks/shortcomings and needed to be addressed. At least for some of these to gain confidence on the study outcome.”
We understand the Reviewer’s position. However, to understand how iron kills neurons, and how HS is protecting, this would require a new series of experiments that would be beyond the scope of this study. For now, we believe that the result that HS protects against iron neurotoxicity should be of interest to the MS field. I hope that the Reviewer accepts this position.
“Line 590. what is the purpose of this paragraph? This is just citing studies and ended with a shortcoming sentence.”
In deference to the Reviewer’s concern, we have now removed this paragraph.
We appreciate the Reviewer for having taken the time to evaluate the manuscript in a critical and thorough manner. We have revised the manuscript as best as we can. We believe that the paper is now ready for publication. We hope the Reviewer agrees. Thank you again.
Round 3
Reviewer 2 Report
I have two comments to address before acceptance.
- Line 81-82. "if other mechanisms noted 81 above of iron-induced injury are prominent". What other mechanisms that authors are referring to?
- Line 83. Replace "is" with "are".
- In figure 2 legend, colors are still not defined.
- Figure6B. Stain colors are not defined.
Author Response
In response to Reviewer #2:
We thank the Reviewer for the prompt review of our revised manuscript, and for the feedback to improve our paper.
“1. Line 81-82. "if other mechanisms noted 81 above of iron-induced injury are prominent". What other mechanisms that authors are referring to?”
Thank you. Now, we have added to that line: “ (e.g. misfolding of proteins)”.
“2. Line 83. Replace "is" with "are".”
“Is” has been replaced with “are”.
“3. In figure 2 legend, colors are still not defined.”
We apologize for this. The blue and red colors in the figure legend have been defined.
“4. Figure6B. Stain colors are not defined.”
We now state in the figure legend that “the respective colors are defined by the words in each panel.”